# EUCLID: Towards Efficient Unsupervised Reinforcement Learning with Multi-choice Dynamics Model

**Yifu Yuan**[1], **Jianye Hao**[*,1], **Fei Ni**[1], **Yao Mu**[3], **Yan Zheng**[1], **Yujing Hu**[2], **Jinyi Liu**[1], **Yingfeng Chen**[2], **Changjie Fan**[2]
[1]College of Intelligence and Computing, Tianjin University,
[2]Fuxi AI Lab, Netease, Inc., Hangzhou, China, [3]The University of Hong Kong

## Abstract

Unsupervised reinforcement learning (URL) poses a promising paradigm to learn useful behaviors in a task-agnostic environment without the guidance of extrinsic rewards to facilitate the fast adaptation of various downstream tasks. Previous works focused on the pre-training in a model-free manner while lacking the study of transition dynamics modeling that leaves a large space for the improvement of sample efficiency in downstream tasks. To this end, we propose an **E**fficient **U**nsupervised reinfor**C**ement **L**earning framework with multi-cho**I**ce **D**ynamics model (**EUCLID**), which introduces a novel model-fused paradigm to jointly pre-train the dynamics model and unsupervised exploration policy in the pre-training phase, thus better leveraging the environmental samples and improving the downstream task sampling efficiency. However, constructing a generalizable model which captures the local dynamics under different behaviors remains a challenging problem. We introduce the multi-choice dynamics model that covers different local dynamics under different behaviors concurrently, which uses different heads to learn the state transition under different behaviors during unsupervised pre-training and selects the most appropriate head for prediction in the downstream task. Experimental results in the manipulation and locomotion domains demonstrate that EUCLID achieves state-of-the-art performance with high sample efficiency, basically solving the state-based URLB benchmark and reaching a mean normalized score of **104.0±1.2**% in downstream tasks with 100k fine-tuning steps, which is equivalent to DDPG's performance at 2M interactive steps with **20×** more data. More visualization videos are released on our [homepage](#).

## 1 Introduction

Reinforcement learning (RL) has shown promising capabilities in many practical scenarios ([Li et al., 2022b](#); [Ni et al., 2021](#); [Shen et al., 2020](#); [Zheng et al., 2019](#)). However, RL typically requires substantial interaction data and task-specific rewards for the policy learning without using any prior knowledge, resulting in low sample efficiency ([Yarats et al., 2021c](#)) and making it hard to generalize quickly to new downstream tasks ([Zhang et al., 2018](#); [Mu et al., 2022](#)). For this, unsupervised reinforcement learning (URL) emerges and suggests a new paradigm: pre-training policies in an unsupervised way, and reusing them as prior for fast adapting to the specific downstream task ([Li et al., 2020](#); [Peng et al., 2022](#); [Seo et al., 2022](#)), shedding a promising way to further promote RL to solve complex real-world problems (filled with various unseen tasks).

Most URL approaches focus on pre-train a policy with diverse skills via exploring the environment guided by the designed unsupervised signal instead of the task-specific reward signal ([Hansen et al., 2020](#); [Liu & Abbeel, 2021a](#)). However, such a pre-training procedure may not always benefit downstream policy learning. As shown in Fig. [1](#), we pre-train a policy for 100k, 500k, 2M steps in the robotic arm control benchmark Jaco, respectively, and use them as the prior for the downstream policy learning to see how pre-training promotes the learning. Surprisingly, long-hour pre-training does not always bring benefits and sometimes deteriorates the downstream learning (500k vs 2M in the

---

*Corresponding authors: Jianye Hao (jianye.hao@tju.edu.cn)

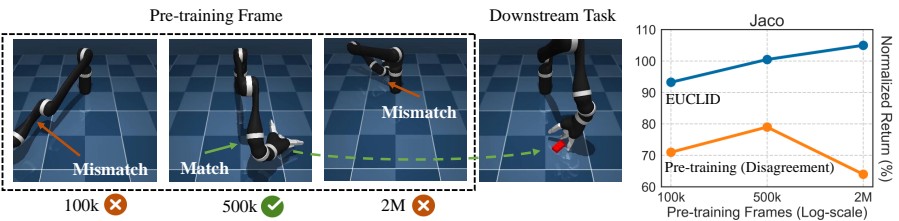

Figure 1: A motivation example.

orange line). We visualize three pre-trained policies on the left of Fig. 1, and find they learn different skills (i.e., each covering a different state space). Evidently, only one policy (pre-trained with 500k) is beneficial for downstream learning as it happens to focus on the area where the red brick exists. This finding reveals that the downstream policy learning could heavily rely on the pre-trained policy, and poses a potential limitation in existing URL approaches: ***Only pre-training policy via diverse exploration is not enough for guaranteeing to facilitate downstream learning***. Specifically, most mainstream URL approaches pre-train the policy in a model-free manner (Pathak et al., 2019; 2017; Campos et al., 2020), meaning that the skill discovered later in the pre-training, will more or less suppress the earlier ones (like the catastrophic forgetting). This could result in an unpredictable skill that is most likely not the one required for solving the downstream task (2M vs. 500k). We refer to this as the *mismatch issue* which could make pre-training even less effective than randomized policy in the downstream learning. Similarly, Laskin et al. (2021) also found that simply increasing pre-training steps sometimes brings no monotonic improvement but oscillation in performance.

To alleviate above issue, we propose the **E**fficient **U**nsupervised reinfor**C**ement **L**earning framework with multi-cho**I**ce **D**ynamic model (**EUCLID**), introducing the model-based RL paradigm to achieve rapid downstream task adaption and higher sample efficiency. First, in the pre-training phase, EUCLID proposes to pre-train the environment dynamics model, which barely suffers from the mismatch issue as the upstream and downstream tasks in most time share the same environment dynamics. Notably, the pre-training dynamics model is also orthogonal to the pre-training policy, thus EUCLID pre-trains them together and achieves the best performance (see Fig. 1). In practice, EUCLID requires merely no additional sampling burden as the transition collected during the policy pre-training can also be used for the dynamics model pre-training. On the other hand, in the fine-tuning phase, EUCLID leverages the pre-trained dynamics model for planning, which is guided by the pre-trained policy. Such a combination could eliminate the negative impact caused by the mismatch issue and gain fast adaptation performance. More importantly, EUCLID can monotonically benefit from an accurate dynamics model through a longer pre-training.

Another practical challenge is that, due to the model capacity, pre-training one single dynamics model is hard to accurately model all the environment dynamics. The inaccuracy can be further exacerbated in complex environments with huge state space, and thus deteriorates the downstream learning performance. Inspired by multi-choice learning, EUCLID proposes a multi-headed dynamics model with each head pre-trained with separate transition data. Each head focuses on a different region of the environment, and is combined to predict the entire environment dynamics accurately. As such, in the fine-tuning phase, EUCLID could select the most appropriate head (sharing a similar dynamics to the downstream task) to achieve a fast adaptation.

Our contributions are four-fold: (1) we extend the mainstream URL paradigm by innovative introducing the dynamics model in the pre-training phase, so that model-based planning can be leveraged in the fine-tuning phase to alleviate the mismatch issue and further boost the downstream policy learning performance; (2) We propose a multi-headed dynamics model to achieve a fine-grained and more accurate prediction, which promotes effective model planning in solving downstream tasks; (3) We empirically study the performance of EUCLID by comparing different mainstream URL mechanisms or designs, and comprehensively analyze how each part of EUCLID affect the ultimate performance; (4) Extensive comparisons on diverse continuous control tasks are conducted and the results demonstrate significant superiority of EUCLID in performance and sample efficiency, especially in challenging environments. Our approach basically solves the state-based URLB, achieving state-of-the-art performance with a normalized score of **104.0±1.2**% and outperforming the prior leading method by **1.35×**, which is equivalent to DDPG with **20×** more data.

## 2 BACKGROUND

**Markov Decision Process (MDP)** is widely used for formulating a continuous control task, defined as a tuple $(\mathcal{S}, \mathcal{A}, R, \mathcal{P}, \rho_0, \gamma)$ of the state space $\mathcal{S}$, action space $\mathcal{A}$, reward function $R(\mathbf{s}, \mathbf{a})$, transition probability $\mathcal{P}(\mathbf{s}' \mid \mathbf{s}, \mathbf{a})$, initial state distribution $\rho_0$ and discounting factor $\gamma$ (Sutton & Barto, 1998). The objective is to learn the optimal policy $\mathbf{a}_t \sim \pi_\phi(\cdot \mid \mathbf{s}_t)$ that maximizes the expected discounted return $\mathbb{E}_{\mathbf{s}_0 \sim \rho_0, (\mathbf{s}_0, \mathbf{a}_0, \dots, \mathbf{s}_T) \sim \pi} \left[ \sum_{t=0}^{T-1} \gamma^t R(\mathbf{s}_t, \mathbf{a}_t) \right]$, where $T$ is the variable episode length.

**Unsupervised reinforcement learning (URL)** poses a promising approach to learning useful priors from past experience and accelerating the learning of downstream tasks (Schwarzer et al., 2021). URLB (Laskin et al., 2021) split the whole learning phase into two parts, pre-training (PT) and fine-tuning (FT).At every timestep in the PT phase, the agents can only interact with the task-agnostic reward-free environment to obtain intrinsic rewards learned through a self-supervised manner. In contrast, in the FT phase, agents need to adapt quickly to downstream tasks with task-specific extrinsic rewards provided by the environment. Specifically, URL algorithms can be generalized into three categories, including knowledge-based, data-based and competence-based methods (Oudeyer et al., 2007; Srinivas & Abbeel, 2021). The goal of the knowledge-based methods is to increase knowledge of the world by maximizing prediction errors (Pathak et al., 2017; 2019; Burda et al., 2019) while data-based methods aim to maximize the entropy of the state of the agents (Liu & Abbeel, 2021b; Yarats et al., 2021a; Hazan et al., 2019). Competence-based methods learn an explicit skill vector by maximizing the mutual information between the observation and skills (Campos et al., 2020; Eysenbach et al., 2019; Gregor et al., 2017).

**Model-based reinforcement learning (MBRL)** leverages a learned dynamic model of the environment to plan a sequence of actions in advance which augment the data (Sutton, 1991; Janner et al., 2019; Pan et al., 2020; Mu et al., 2020; Peng et al., 2021) or obtain the desired behavior through planning (Chua et al., 2018; Hafner et al., 2019; Lowrey et al., 2019; Mu et al., 2021; Chen et al., 2022). However, training the world model requires a large number of samples (Polydoros & Nalpantidis, 2017; Plaat et al., 2021), and an imprecise model can lead to low-quality decisions for imaginary planning (Freeman et al., 2019). Thus constructing a reliable world model by pre-training can greatly accelerate the learning process of downstream tasks (Chebotar et al., 2021; Seo et al., 2022). A concurrent work of ours is Rajeswar et al. (2022), which also focuses on promoting URL performance through a world model, but only considers the incorporation of a simple single dynamics model without realizing the low accuracy caused by the mismatch of the pre-training policies and the optimal policies for the downstream tasks. Our work designs multi-choice learning and policy constraints to explore how to construct a generic pre-trained dynamics model to cover different local dynamics under different behaviors.

**Multi-choice learning (MCL)** learns ensemble models that produce multiple independent but diverse predictions and selects the most accurate predictions to optimize the model, better reduce training loss and improve stability (Guzmán-Rivera et al., 2012; 2014; Dey et al., 2015; Lee et al., 2017). Multi-choice learning has been widely used, such as in vision computing (Cuevas et al., 2011; Tian et al., 2019; Lee et al., 2017) and visual question answer (Lei et al., 2020; Lu et al., 2022), but most works focus on supervised learning. T-MCL (Seo et al., 2020) and DOMINO (Mu et al., 2022) bring MCL into the meta RL, which approximates multi-modal distribution with context and models the dynamics of changing environments. Inspired by this, EUCLID builds a multi-headed dynamics model to cover specialized prediction regions corresponding to different downstream tasks.

## 3 METHODOLOGY

In this work, we propose an **E**fficient **U**nsupervised reinfor**C**ement **L**earning framework with multi-choI**ce** **D**ynamic model (**EUCLID**) to further improve the ability of mainstream URL paradigm in fast adapting to various downstream tasks. As a starter, EUCLID adopts the task-oriented latent dynamic model as the backbone for environment modeling, and contains two key parts: ❶ a model-fused URL paradigm that innovatively integrates the world model into the pre-training and fine-tuning for facilitating downstream tasks learning, and ❷ a multi-headed dynamics model that captures different environment dynamics separately for an accurate prediction of the entire environment. In this way, EUCLID can achieve a fast downstream tasks adaptation by leveraging the accurate pre-trained environment model for an effective model planning in the downstream fine-tuning. The detail pseudo code is given in Algorithm 1.

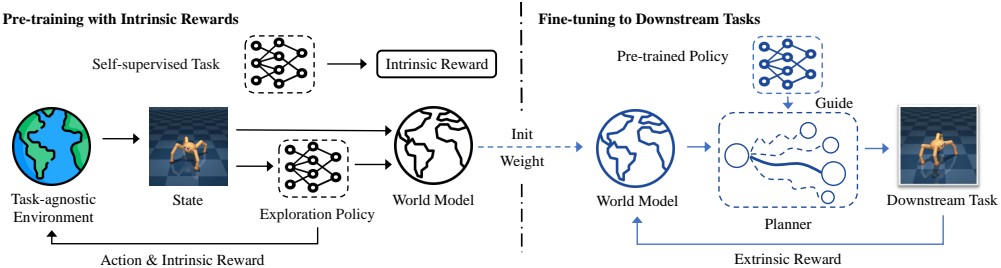

Figure 2: **Overview of the model-fused URL paradigm**. In the pre-training phase (left), we jointly update the world models and policy using exploring samples collected from the environment interaction. In the fine-tuning phase (right), we reuse the pre-trained weights to initialize the downstream world models and the policy, perform policy guided planning via dynamics model for fast adaption.

### 3.1 BACKBONE FOR ENVIRONMENT MODELING

We build world models for representation learning and planning based on the Task-Oriented Latent Dynamic Model (TOLD) of TDMPC (Hansen et al., 2022). EUCLID learns world models that compresses the history of observations into a compact feature space variable $\mathbf{z}$ and enables policy learning and planning in this space (Zhang et al., 2019; Hafner et al., 2019). EUCLID's world model components include three major model components: (i) the representation model that encodes state $\mathbf{s_t}$ to a model state $\mathbf{z_t}$ and characterizes spatial constraints by latent consistency loss, (ii) the latent dynamics model that predicts future model states $\mathbf{z}_{t+1}$ without reconstruction and (iii) the reward predictor allows the model to encoder task-relevant information in a compact potential space. The model can be summarized as follow:

$$\text{Representation:} \quad \mathbf{z}_t = E_\theta\left(\mathbf{s}_t\right) \qquad \text{Latent dynamics: } \mathbf{z}_{t+1} = D_\theta\left(\mathbf{z}_t, \mathbf{a}_t\right)$$
$$\text{Reward predictor:} \quad \hat{r}_t \sim R_\theta\left(\mathbf{z}_t, \mathbf{a}_t\right) \tag{1}$$

where $\mathbf{s}, \mathbf{a}, \mathbf{z}$ denote a state, action and latent state representation. Then, for effective value-based learning guidance planning, we built actor and critic based on DDPG (Lillicrap et al., 2016):

$$\text{Value: } \hat{q}_t = Q_\theta\left(\mathbf{z}_t, \mathbf{a}_t\right) \qquad \text{Policy: } \hat{\mathbf{a}}_t \sim \pi_\phi\left(\mathbf{z}_t\right) \ , \tag{2}$$

All model parameters $\theta$ except actor are jointly optimized by minimizing the temporally objective:

$$\mathcal{L}\left(\theta, \mathcal{D}\right) = \mathbb{E}_{(\mathbf{s}, \mathbf{a}, \mathbf{z}_t, r_t)_{t:t+H} \sim \mathcal{D}}\left[\sum_{i=t}^{t+H}\left(c_1 \underbrace{\left\|R_\theta\left(\mathbf{z}_i, \mathbf{a}_i\right) - r_i\right\|_2^2}_{\text{Reward prediction}} + c_2 \underbrace{\left\|D_\theta\left(\mathbf{z}_i, \mathbf{a}_i\right) - E_{\theta^-}\left(\mathbf{s}_{i+1}\right)\right\|_2^2}_{\text{Latent state consistency}}\right.\right.$$
$$\left.\left. + c_3 \underbrace{\left\|Q_\theta\left(\mathbf{z}_i, \mathbf{a}_i\right) - \left(r_i + \gamma Q_{\theta^-}\left(\mathbf{z}_{i+1}, \pi_\theta\left(\mathbf{z}_{i+1}\right)\right)\right)\right\|_2^2}_{\text{Value prediction}}\right)\right] , \tag{3}$$

while actor $\pi_\phi$ maximizes $Q_\theta$ approximated cumulative discounted returns. The reward term predicts a single-step task reward, the state transition term aims to predict future latent states accurately and the value term is optimized through TD-learning (Haarnoja et al., 2018; Lillicrap et al., 2016). Hyper-parameters $c_i$ are adjusted to balance the multi-source loss function and a trajectory of length $H$ is sampled from the replay buffer $\mathcal{D}$.

### 3.2 MODEL-FUSED UNSUPERVISED REINFORCEMENT LEARNING PARADIGM

Overall, EUCLID introduces the environment modeling into the pre-training (PT) and fine-tuning (FT) phases, formulating a new model-fused URL paradigm. As shown in Fig. 2 (left), EUCLID adopts a reward-free exploration policy to collect task-agnostic transitions for pre-training the dynamics model and then uses them to boost downstream FT. In the following, we describe three important detail designs in EUCLID throughout the PT and FT processes.

Firstly, in the PT phase, we expect as diverse data as possible to pre-train the dynamics model for an accurate environment prediction. As EUCLID can be easily combined with almost any mainstream exploration methods (Hao et al., 2023), we empirically evaluate the knowledge-based method (Disagreement, Pathak et al. (2019)), data-based method (APT, Liu & Abbeel (2021b)) and competence-based method (DIAYN, Eysenbach et al. (2019)) (see more in Appendix B). We find

Figure 3: **Illustrations of multi-choice learning in EUCLID**. We design a multi-headed dynamics model to predict different environment dynamics, training each predictor head separately using diverse data, and select the most beneficial predictor head for the downstream task.

the knowledge-based method (i.e., Disagreement) synthetically performs the best, and hypothesize this happens because it explores via maximizing the prediction uncertainty, allowing discovery more unpredictable dynamics.

Secondly, in the PT phase, how to select the loss function for the dynamics model pre-training is challenging. We use latent consistency loss (Schwarzer et al., 2020; 2021; Hansen et al., 2022) to learn the dynamics model in latent space to directly predict the feature of future states without reconstruction (Hafner et al., 2020b;a; Ha & Schmidhuber, 2018). This improves the generalization of the dynamics model and avoids capturing task-irrelevant details between PT and FT phase while the reconstruction loss forces to model everything in the environment in such a huge state space.

Lastly, shown in Fig. 2 (right), we utilize the pre-trained dynamics model to rollout trajectories for both planning and policy gradient in the FT for downstream tasks. An important reason is that model-based planning can yield a stable and efficient performance due to the similarity of the environment dynamics between upstream and downstream tasks, thus an accurate pre-trained dynamics model could benefit downstream learning. This can avoid the above-mentioned mismatch issue, caused by utilizing a less effective pre-trained policy for the downstream tasks. For this, we adopt a representative model predictive control method (Williams et al., 2015) to select the best action based on the imaged rollouts (via dynamics model). Meanwhile, we also notice that the pre-trained policy can master some simple skills like rolling, swinging, and crawling, which is useful and can be utilized in the model planning to further boost downstream learning. Therefore, we additionally mix the trajectories generated by the pre-trained policy interacting with the dynamics model for planning. This would further speed up the planning performance and effectiveness, especially in the early stage of fine-tuning (Wang & Ba, 2020). Another side benefit is that the $Q$-value function (in the critic) in the pre-trained policy could speed up the long-term planning via bootstrapping (Sikchi et al., 2021) (see details in Appendix H).

### 3.3 MULTI-CHOICE LEARNING VIA MULTI-HEADED DYNAMICS MODEL

Prior works (Wu et al., 2019; Seo et al., 2020) reveal that training one single dynamics model to accurately predict the whole environment dynamics is difficult in complex environments. Besides, pre-training one single dynamics model to predict the entire state space in the environment is ineffective, as downstream tasks may involve only limited local state distributions (left in Fig. 3). Therefore, EUCLID proposes a multi-choice learning mechanism via multi-headed dynamics model in the PT stage for achieving a more accurate environment prediction and to select one of the heads for the FT stage to further boost downstream learning.

As shown in Fig. 3 (middle), we design a multi-headed dynamics model to predict different environment dynamics (i.e., state distributions), encouraging each predictor head to enjoy better prediction accuracy in its own specialized prediction region. The model contains a backbone parameterized by $\theta$, sharing the high-level representations of the first few layers of the neural network, while the latter layers build $H$ prediction heads parameterized by $\{\theta_h^{\text{head}}\}_{h=1}^{H}$. The output is:

$$\mathbf{z}_{t+1} = \left\{ D_\theta \left( \mathbf{z}_t, a_t; \theta, \theta_h^{\text{head}} \right) \right\}_{h=1}^{H}. \tag{4}$$

We do not use separate ensemble models with completely different parameters because we empirically observe that there may also be a large number of shared dynamics between different downstream tasks which can be captured by the shared backbone to leverage environmental samples efficiently and also avoid actively reducing the information of acquired environment samples.

To encourage each head to focus on a separate region as possible, we assign different exploration policies for each head, denoted by $[\pi_\phi^1(\mathbf{z}), \cdots, \pi_\phi^h(\mathbf{z})]$. To avoid region overlap, we design a diversity-encouraging term as an additional regularization term as follows:

$$\mathcal{L}_\pi(\phi, \mathcal{D}) = \mathbb{E}_{\mathbf{z}_t \sim \mathcal{D}} \left[ Q_\theta\left(\mathbf{z}_t, \pi_\phi\left(\mathbf{z}_t\right)\right) - \alpha D_{\mathrm{KL}}\left(\widetilde{\pi}_\phi\left(\mathbf{z}_t\right) \| \pi_\phi\left(\mathbf{z}_t\right)\right)\right], \tag{5}$$

where $\widetilde{\pi}_\phi(\mathbf{z}_t) = \sum_{i=1}^h \pi_i(\mathbf{z})/h$. In this way, each policy is optimized by its own intrinsic for exploration and policies are encouraged to move away from each other (see more in Appendix G).

In the beginning of FT phase, we select one head in the pre-trained multi-headed dynamics model to construct a single-head dynamic model that can benefit the downstream learning the most (right in Fig. 3). To figure out the optimal head $h^*$ (that covers the most appropriate state region) for solving the downstream tasks, inspired by Seo et al. (2020), we evaluate the zero-shot performance of each head and pick one with the highest zero-shot performance in downstream tasks. After that, only optimal head is used in the subsequent fine-tuning for adapting to downstream tasks.

Overall, we empirically show that different prediction heads can indeed cover different state regions with respective closely related tasks and the multi-choice learning improves the performance by selecting the appropriate specialized prediction head (see Fig. 7).

## 4 EXPERIMENTS

We conduct experiments on various tasks to study the following research questions (RQs):
**Combination (RQ1):** Which category of URL method works best in combination with EUCLID?
**Performance (RQ2):** How does EUCLID compare to existing URL methods?
**Monotonism (RQ3):** Can EUCLID monotonically benefit from longer pre-training steps?
**Specialization (RQ4):** Does multi-choice learning delineate multiple specialized prediction region?
**Ablation (RQ5):** How each module in EUCLID facilitates the downstream policy learning?

### 4.1 EXPERIMENTAL SETUP

**Benchmarks:** We evaluate our approach on tasks from URLB (Laskin et al., 2021), which consists of three domains (walker, quadruped, jaco) and twelve challenging continuous control downstream tasks. Besides, we extend the URLB benchmark (URLB-Extension) by adding a more complex humanoid domain and three corresponding downstream tasks based on the DeepMind Control Suite (DMC) (Tunyasuvunakool et al., 2020) to further demonstrate the efficiency improvement of EUCLID on more challenging environments. Environment details can be found in Appendix A.

**Baselines:** The proposed EUCLID is a general framework that can be easily combined with any unsupervised RL algorithms. Therefore we make a comprehensive investigation of EUCLID combined with popular unsupervised RL algorithms of three exploration paradigms. Specifically, we choose the most representative algorithms from each of the three categories including Disagreement (Knowledge-based) (Pathak et al., 2019), APT (Data-based) (Liu & Abbeel, 2021b) and DIAYN (Competence-based) (Eysenbach et al., 2019). In addition, we compare our method to the previous state-of-the-art method CIC (Laskin et al., 2022) in the URLB, which is a hybrid data-based and competence-based method. More detailed information about the baselines is in Appendix B.

**Evaluation:** In the PT phase, all model components are pre-trained under an unsupervised setting based on the sample collected by interacting 2M steps in the reward-free environment. Then in the FT phase, we fine-tune each agent in downstream tasks with the extrinsic reward for only 100k steps, which is moderated to 150k steps for the humanoid domain due to special difficulties. All statistics are obtained based on 5 independent runs per downstream task, and we report the average with 95% confidence regions. For fairness of comparison, all model-free baselines and the policy optimization part of model-based methods opt DDPG (Lillicrap et al., 2016) agent as backbone, maintaining the same setting as Laskin et al. (2021). To better compare the average performance of multiple downstream tasks, we choose the expert score of DDPG learning from scratch with 2M steps in the URLB as a normalized score. Note that expert agents run for 2M steps with extrinsic reward, which is **20×** more than the budget of the agent used for evaluation steps in the FT phase.

**Implementation details:** EUCLID employs both model planning and policy optimization in the FT phase but uses only policy optimization in the PT phase. During planning, we leverage Model Predictive Path Integral (MPPI) (Williams et al., 2015) control and choose a planning horizon of

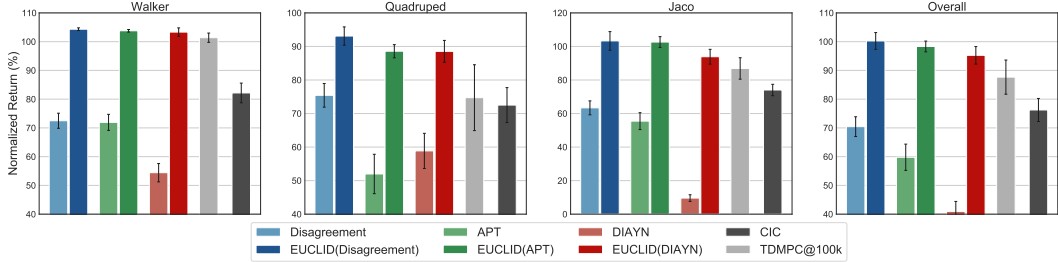

Figure 4: We combine EUCLID with three types of exploration algorithms to show the synergistic benefit of URL and MBRL methods for 12 downstream tasks on URLB. The normalized scores are the average performance after pre-training 2M steps and then fine-tuning 100k steps with 5 independent seeds. EUCLID achieves significant improvement over the corresponding exploration backbone and empirically shows the combination with the knowledge-based method is most effective.

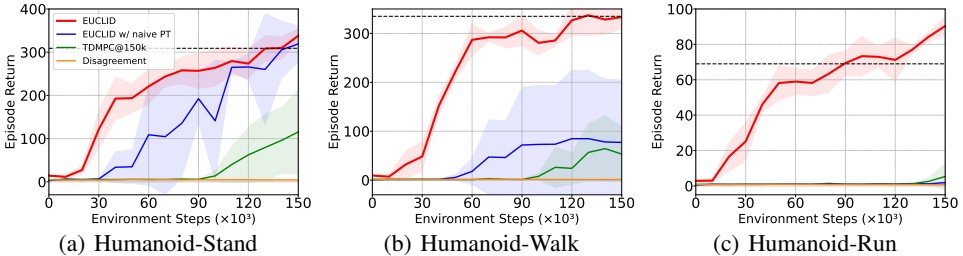

| (a) Humanoid-Stand | (b) Humanoid-Walk | (c) Humanoid-Run |

Figure 5: Learning curves of EUCLID and three baselines on downstream tasks of the humanoid domain. Curves show the mean and 95% confidence intervals of performance across 5 independent seeds. The dashed reference lines are the asymptotic performance of the Disagreement algorithm with 2M PT steps and 2M FT steps. These results show that EUCLID has better sample efficiency and large performance gains on tasks with complex dynamics.

$L = 5$. Notably, we still retain seed steps for warm-start in the FT phase despite sufficient pre-training steps to alleviate the extreme case of a strong mismatch between the pre-training policies and the optimal policies for the downstream tasks. For multi-choice learning, we build a multi-headed dynamics model maintaining heads of $H = 4$ by default, each optimized by independent samples. To avoid unfair comparison, we keep the same training settings for all methods.

## 4.2 COMBINATION (RQ1)

To answer RQ1, we show the synergistic benefit of EUCLID in unsupervised RL methods and model-based architecture. As shown in Fig. 4, We build EUCLID on a different URL backbone named EUCLID (Disagreement/APT/DIAYN), where the dark line denotes its normalized score and the light line denotes the corresponding vanilla URL backbone results in Laskin et al. (2021). For the purpose of simplicity, *we do not use the multi-choice learning mechanism here*. We show that we can combine all three types of exploration baselines and obtain significant performance gains in all environments. In particular, the previous competence-based approaches under-perform in URLB because of the weak discriminator, while the EUCLID (DIAYN) greatly improves fast adaptation to downstream tasks through pre-trained world models. Besides, we empirically find that EUCLID is better combined with the knowledge-based approaches with the best performance and stability. Unless specified, we choose Disagreement as the exploration backbone in the PT stage by default.

## 4.3 PERFORMANCE (RQ2)

**Comparative evaluation on URLB.** To answer RQ2, we evaluate the performance of EUCLID and other baseline methods in the URLB and URLB-Extension environment. The results in Table 1 show that EUCLID outperforms all other mainstream baselines in all domains and basically solves the twelve downstream tasks of the state-based URLB in 100k steps, obtaining a performance comparable to the expert scores while the expert agent trained 2M steps with extrinsic reward. It is worth noting that EUCLID significantly improves performance in reward-sparse robotic arm control tasks, such as jaco, which is challenging for previous URL methods. Moreover, the results of EU-

Table 1: Performance of EUCLID and EUCLID w/o multi headed structure on URLB after 2M reward-free pre-training steps and finetuning for 100k steps with extrinsic rewards. All baseline runs with 5 seeds. We refer to Appendix C for full results of all baselines.

| Domain | Task | Disagreement | CIC | TDMPC@100k | EUCLID w/o MCL | EUCLID |
|--------|------|--------------|-----|------------|----------------|--------|
| Walker | Flip | 491±21 | 631±34 | 930±28 | **971±1** | **969±2** |
| | Run | 444±21 | 486±25 | 750±4 | **765±10** | **770±9** |
| | Stand | 907±15 | 959±2 | 940±22 | **985±1** | **985±1** |
| | Walk | 782±33 | 885±28 | 967±2 | **967±4** | **972±1** |
| | Normalized score | 72.5±2.6 | 82.2±3.4 | 101.4±1.6 | **104.3±0.5** | **104.6±0.4** |
| Quadruped | Jump | 668±24 | 595±42 | 723±98 | 840±13 | **858±14** |
| | Run | 461±12 | 505±47 | 465±83 | 651±35 | **735±16** |
| | Stand | 840±33 | 761±54 | 765±92 | **953±6** | **958±5** |
| | Walk | 721±56 | 723±43 | 710±77 | 874±42 | **925±6** |
| | Normalized score | 73.5±3.5 | 72.5±5.2 | 74.7±9.8 | 93.1±2.7 | **97.6±1.1** |
| Jaco | Reach bottom left | 134±8 | 138±9 | 168±7 | 214±5 | **220±3** |
| | Reach bottom right | 122±4 | 145±7 | 183±11 | 205±9 | **212±2** |
| | Reach top left | 117±14 | 153±7 | 178±11 | 197±23 | **225±5** |
| | Reach top right | 140±47 | 163±4 | 172±24 | 219±7 | **229±6** |
| | Normalized score | 63.4±8.6 | 74.0±3.4 | 86.9±6.4 | 103.3±5.6 | **109.7±2.0** |
| Overall | Normalized score | 69.8±4.9 | 76.2±4.0 | 87.7±6.0 | 100.2±2.9 | **104.0±1.2** |

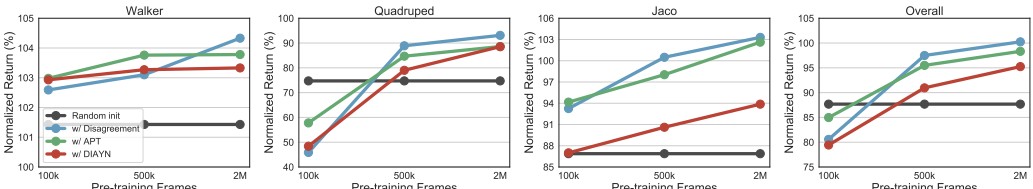

Figure 6: We show the fine-tuning performance of EUCLID with different pre-training steps, including 100k, 500k, and 2M steps. EUCLID can benefit from longer pre-training steps to achieve monotonic performance gains, while previous URL algorithms fail.

CLID also show significant improvement over TDMPC@100k learned from scratch, especially with greater gains in the Quadruped domain where is challenging to predict accurately. This suggests that unsupervised pre-training can significantly reduce model error.

**Effects of multi-headed dynamics model.** To verify the importance of multi-choice learning, we compared the performance of EUCLID and EUCLID w/o multi-headed dynamics model. Table 1 demonstrates that EUCLID achieved almost all of the 12 downstream tasks' average performance improvement and significantly reduced the variance, reaching the highest normalized score of **104.0±1.2**%. This indicates that multi-choice learning reduces the prediction error of the downstream tasks corresponding to local dynamics and obtains better results.

**URLB-Extension experiments.** As shown in Fig. 5, we evaluate the agents in the humanoid domain which is the most difficult to learn locomotion skills due to high-dimensional action spaces. We found that the previous model-free URL method is difficult to improve the learning efficiency of downstream tasks in such a complex environment, while the previous model-based methods (TDMPC) still make some initial learning progress. In contrast, EUCLID is able to consistently improve performance in all three downstream tasks with small variance. Also, we compare EUCLID with the naive pre-training scheme, which uses a random exploration policy to collect data in the PT phase to train the world models. As the task difficulty increases, we observe that EUCLID w/ naive PT struggles to achieve competitive performance because of the lack of intrinsic unsupervised goal.

## 4.4 MONOTONISM (RQ3)

Intuitively, a longer pre-training phase is expected to allow for a more efficient fine-tuning phase, but the empirical evidence of Laskin et al. (2021) demonstrated that longer pre-training is not always beneficial or even worse than random initialization when applying the previous URL algorithm, because it is difficult to capture such rich environmental information with only pre-trained policies. This is a major drawback that makes the application URL pre-training paradigm fraught with uncertainty. However, EUCLID better leverage environmental information to build additional world models. Fig. 6 shows that our methods achieve monotonically improving performance by varying pre-training steps on most of the tasks. All results are better than random initialization, except for the quadruped domain with 100k pre-training steps.

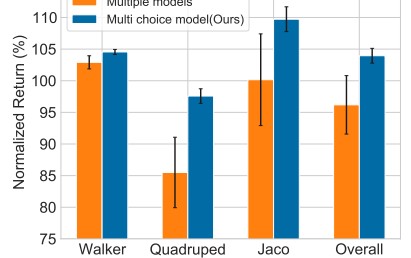

| Walker | Head 1 | Head 2 | Head 3 | Head 4 |
|---|---|---|---|---|
| Flip | 267 | 266 | 183 | 268 |
| Run | 66 | 105 | 109 | 100 |
| Stand | 419 | 408 | 388 | 422 |
| Walk | 178 | 239 | 218 | 186 |

| Quadruped | Head 1 | Head 2 | Head 3 | Head 4 |
|---|---|---|---|---|
| Jump | 296 | 324 | 242 | 369 |
| Run | 215 | 233 | 138 | 228 |
| Stand | 431 | 323 | 359 | 353 |
| Walk | 217 | 166 | 173 | 161 |

(a) Specialized region of each predict head  (b) Importance of multi-headed model

Figure 7: (a) Zero-shot performance of the pre-trained multi-headed dynamics model for different tasks on Walker and Quadruped domain. We highlight the top-2 prediction heads which are suitable for Corresponding tasks. (b) The average performance of the multi-headed dynamics model used by EUCLID and multiple dynamics models across 12 downstream tasks for 2M pre-training steps.

## 4.5 SPECIALIZATION (RQ4)

**Prediction heads corresponding to the specialized regions.** To investigate the ability of EU-CLID to learn specialized prediction heads, we visualized how to assign heads to downstream tasks in Fig. 7(a), we can observe that different prediction heads are better adapted to different downstream tasks, with their corresponding specialized region. i.e., head 2 works best for Run tasks while head 4 for Jump tasks in the Quadruped environment. No one prediction head can perform optimally for all downstream tasks. See our homepage for more visualization analysis.

**Comparison of multiple models and multi-headed models.** Fig. 7(b) shows the comparison of the multi-headed dynamics model used by EUCLID and multiple dynamics models without a shared backbone. We find that multiple models are less effective than multi-headed models in all domains, especially in the quadruped domain which requires a large amount of sample to model complex dynamics. This also demonstrates that sharing the underlying dynamics of agents is beneficial to improve the sample efficiency for downstream tasks.

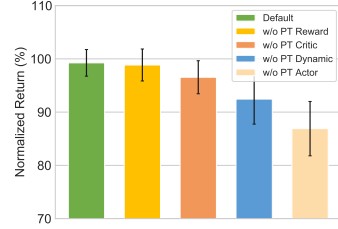

Figure 8: Ablation on different pre-trained components for FT.

## 4.6 ABLATION (RQ5)

**Roles for each module.** As shown in Fig. 8, We conducted ablation experiments with 2M steps pretraining on the EUCLID, reusing different subsets of pre-trained components in the FT phase. To ensure the generality of the conclusions, we conduct experiments based on different URL methods and take the average results. From the overall results, our default settings, reusing all components of the world models (encoder, dynamics model, reward model) and policy (actor and critic) work best. We show the detailed results for each domain in Appendix E with in-depth analysis.

## 5 CONCLUSION

In this work, we propose EUCLID which introduces the MBRL methods into the URL domain to boost the fast adaption and learning performance. We formulate a novel model-fused URL paradigm to alleviate the mismatch issue between the upstream and downstream tasks and propose the multi-choice learning mechanism for the dynamics model to achieve more accurate predictions and further enhance downstream learning. The results demonstrate that EUCLID achieves state-of-the-art performance with high sample efficiency. Our framework points to a novel and promising paradigm for URL to improve the sample efficiency and may be further improved by the more fine-grained dynamics divided by skill prior (Pertsch et al., 2020), or the combination of offline RL (Yu et al., 2022). In addition, extending EUCLID in the multiagent RL (Li et al., 2022a; Hao et al., 2022; Zheng et al., 2018; 2021) is also a promising direction, which we leave as future work.

ACKNOWLEDGMENTS

This work is supported by the National Natural Science Foundation of China (Grant No.62106172), the "New Generation of Artificial Intelligence" Major Project of Science & Technology 2030 (Grant No.2022ZD0116402), and the Science and Technology on Information Systems Engineering Laboratory (Grant No.WDZC20235250409, No.WDZC20205250407).

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

## A   ENVIRONMENT DETAILS

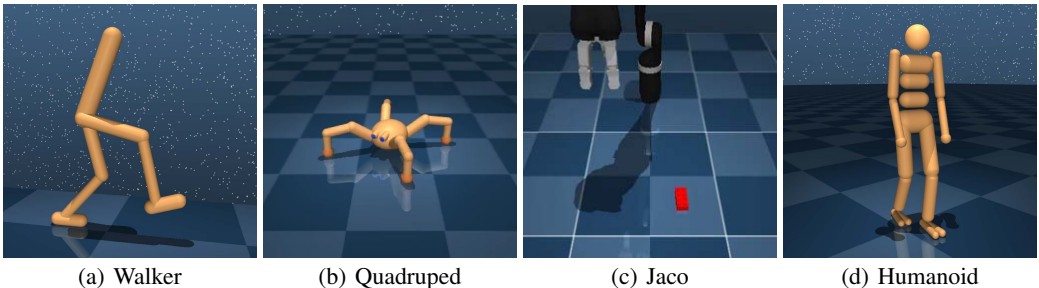

| (a) Walker | (b) Quadruped | (c) Jaco | (d) Humanoid |

Figure 9: **Environment Details.** We evaluate our method on four manipulation and locomotion domains. Each domain contains 4 downstream tasks (3 tasks for humanoid).

The details of all environments and corresponding tasks are as follows, as illustrated in Fig. 9:

- **Walker**[1] *(Flip, Run, Stand, Walk)*: Walker is a biped bound to a 2D vertical plane, which learns balancing and locomotion skills.
- **Quadruped** *(Jump, Run, Stand, Walk)*: Quadruped also learns various types of locomotion skills but is harder because of high-dimensional state and action spaces and 3D environment.
- **Jaco** *(Reach bottom left, Reach bottom right, Reach top left, Reach top right)*: Jaco Arm is a 6-DOF robotic arm with a three-finger gripper, which requires to control the robot arm and perform simple manipulation tasks. Prior works (Laskin et al., 2021; Yarats et al., 2021a) have shown that such reward-sparse manipulation tasks are particularly challenging for unsupervised RL.
- **Humanoid**[2] *(Run, Stand, Walk)*: Humanoid is a simplified human-like entity with 21 joints, which is extremely challenging to learn locomotion skills due to high-dimensional action spaces and exploration dilemma. We test fine-tuning performance on relatively few interaction steps (150k) whereas prior work typically executes for 30M steps (200x) learning from scratch.

## B   DETAILS OF UNSUPERVISED RL BASELINES

EUCLID can simply combine with various categories of URL exploration approaches as backbone. We select several typical exploration algorithms as baselines.

**Disagreement** (Pathak et al., 2019): Disagreement is a knowledge-based baseline which trains an ensemble of forward models $\{g_i (\mathbf{z}_{t+1} \mid \mathbf{z}_t, \mathbf{a}_t)\}$ to predict the feature. Intrinsic rewards are defined as the variance among the ensemble models:

$$r_t^{\text{Disagreement}} \propto \text{Var}\left\{g_i\left(\mathbf{z}_{t+1} \mid \mathbf{z}_t, \mathbf{a}_t\right)\right\} \quad i = 1, \ldots, N. \tag{6}$$

**APT** (Liu & Abbeel, 2021b): Active Pre-training (APT) is a data-based baseline which utilizes a particle-based estimator (Singh et al., 2003) that uses K nearest-neighbors to estimate entropy for a given state. Since APT needs the auxiliary representation learning to provide the latent variables for entropy estimation, we implement of APT on the top of EUCLID encoder optimized by latent consistency loss. Intrinsic reward are computed as:

$$r_t^{\text{APT}} \propto \sum_i^k \log \|\mathbf{z}_t - \mathbf{z}_k\|^2. \tag{7}$$

---

[1] https://github.com/rll-research/url_benchmark
[2] https://github.com/deepmind/dm_control/blob/main/dm_control/suite/humanoid.py

**DIAYN** (Eysenbach et al., 2019): Diversity is All you need (DIAYN) is a competence-based base-line which learn explicit skill $\mathbf{w}$ and maximizes the mutual information between states and skills $I(\mathbf{w}_t, \mathbf{z}_t)$. DIAYN decompose the mutual information via $I(\mathbf{z}_t; \mathbf{w}_t) = H(\mathbf{w}_t) - H(\mathbf{w}_t \mid \mathbf{z}_t)$. The first term is sampled by a random prior distribution $H(\mathbf{w}_t)$ and maximizes the entropy while the latter term is estimated by the discriminator $\log q(\mathbf{w}_t \mid \mathbf{z}_t)$. Intrinsic reward are computed as:

$$r_t^{\text{DIAYN}} \propto \log q(\mathbf{w}_t \mid \mathbf{z}_t) + \text{ const.} \tag{8}$$

**CIC** (Laskin et al., 2022): CIC is a hybrid data-based and competence-based as the previous state-of-the-art method. CIC that uses an intrinsic reward structure based on particle entropy similar to APT and distill behaviors into skills using contrastive learning.

## C    FULL RESULTS ON THE URLB

The full results of fine-tuning for 100k frames for each task and each method are presented in Table 2 on the state-based URLB. For Disagreement, APT and DIAYN, we utilize the results presented in this Laskin et al. (2021).

Table 2: Full results of pre-training for 2M and fine-tuning for 100k steps on the state-based URLB.

| Domain | Task | Disagreement | APT | DIAYN | CIC | TDMPC@100k | EUCLID (Disagreement) w/o MCL | EUCLID (APT) w/o MCL | EUCLID (DIAYN) w/o MCL | EUCLID |
|---|---|---|---|---|---|---|---|---|---|---|
| walker | Flip | 491±21 | 477±16 | 381±17 | 631±34 | 930±28 | 971±1 | 967±3 | 923±34 | **969±2** |
| | Run | 444±21 | 344±28 | 242±11 | 486±25 | 750±4 | 765±10 | 744±9 | 793±7 | **770±9** |
| | Stand | 907±15 | 914±8 | 860±26 | 959±2 | 940±22 | 985±1 | 987±1 | 970±5 | **985±1** |
| | Walk | 782±33 | 759±35 | 661±26 | 885±28 | 967±2 | 967±4 | 974±2 | 967±2 | **972±1** |
| Quadruped | Jump | 668±24 | 462±48 | 578±46 | 595±42 | 723±98 | 840±13 | 792±5 | 807±34 | **858±14** |
| | Run | 461±12 | 339±40 | 415±28 | 505±47 | 465±83 | 651±35 | 589±13 | 571±19 | **735±16** |
| | Stand | 840±33 | 622±57 | 706±48 | 761±54 | 765±92 | 953±6 | 911±31 | 918±30 | **958±5** |
| | Walk | 721±56 | 434±64 | 406±64 | 723±43 | 710±77 | 874±42 | 864±21 | 859±33 | **925±6** |
| Jaco | Reach bottom left | 134±8 | 88±12 | 17±5 | 138±9 | 168±7 | 214±5 | 207±6 | 190±10 | **220±3** |
| | Reach bottom right | 122±4 | 115±12 | 31±4 | 145±7 | 183±11 | 205±9 | 213±6 | 179±11 | **212±2** |
| | Reach top left | 117±14 | 112±11 | 11±3 | 153±7 | 178±11 | 197±23 | 204±7 | 192±10 | **225±5** |
| | Reach top right | 140±47 | 136±5 | 19±4 | 163±4 | 172±24 | 219±7 | 204±7 | 197±4 | **229±6** |

In Table 2, we find that the best approach is to use EUCLID with multi-choice learning and with disagreement as the backbone (EUCLID (Disagreement) with MCL) and we refer to EUCLID for this default setting. EUCLID versions constructed based on each of the explored approaches significantly improve the performance of the corresponding baseline.

## D    IMPLEMENTATION DETAILS

We provide richer implementation details of EUCLID in this section.

### D.1    DEEP DETERMINISTIC POLICY GRADIENT (DDPG)

For continuous action space, we opt for DDPG (Lillicrap et al., 2016) as our base optimization algorithm. DDPG is an actor-critic off-policy algorithm, which update critics $Q_\theta$ by minimizing the Bellman error:

$$\mathcal{L}_Q(\theta, \mathcal{D}) = \mathbb{E}_{(\mathbf{s}_t, \mathbf{a}_t, r_t, \mathbf{s}_{t+1}) \sim \mathcal{D}} \left[ \left( Q_\theta \left( \mathbf{s}_t, \mathbf{a}_t \right) - y(\mathbf{s}) \right)^2 \right], \tag{9}$$

where the Q-target $y(s) = R(s, \mathbf{a}) + \gamma Q_{\bar{\theta}} \left( \mathbf{s}_{t+1}, \pi_\phi \left( \mathbf{s}_{t+1} \right) \right)$, $\mathcal{D}$ is a replay buffer and $\bar{\theta}$ is an slow-moving average of the online critic parameters. At the same time, we learn a policy $\pi_\phi$ that maximizes $Q_\theta$ by maximizing the objective:

$$\mathcal{L}_\pi(\phi, \mathcal{D}) = \mathbb{E}_{\mathbf{s}_t \sim \mathcal{D}} \left[ Q_\theta \left( \mathbf{s}_t, \pi_\phi \left( \mathbf{s}_t \right) \right) \right]. \tag{10}$$

### D.2    HYPER-PARAMETERS

EUCLID studies various categories of exploration algorithms as a backbone in the experiments, and we list the individual hyper-parameters of each method in Table 3. As for the world models and policy, most of the parameters remain the same as in the original TOLD of TDMPC (Hansen et al.,

Table 3: Hyper-parameters of the exploration algorithms used in our experiments.

| Disagreement hyper-parameter | Value |
|---|---|
| Ensemble size | 5 |
| Forward net | $(|\mathcal{O}| + |\mathcal{A}|) \rightarrow 1024$ $\rightarrow 1024 \rightarrow |\mathcal{O}|$ ReLU MLP |
| **APT hyper-parameter** | **Value** |
| Representation dim | 1024 |
| Reward transition | $log(r + 1.0)$ |
| $k$ in NN | 12 |
| Avg top $k$ in NN | True |
| **DIAYN hyper-parameter** | **Value** |
| Skill dim | 16 |
| Skill sampling frequency | 50 |
| Discriminator net | $512 \rightarrow 1024$ $\rightarrow 1024 \rightarrow 16$ ReLU MLP |

2022). We list the hyper-parameters of EUCLID in Table 4, with particular reference to the fact that some of them change during the pre-training phase (PT) and the fine-tuning phase (FT). Following prior work (Hafner et al., 2020b; Hansen et al., 2022), we use a task-specific action repeat hyper-parameter for URLB based on DMControl, which is set to 2 by default while 4 for the Quadruped domain. During the pre-training process, we increase the number of policies and prediction heads of the dynamics model one after another based on specific snapshot time steps, the detailed parameters are shown in the Table 5.

### D.3 COMPUTE RESOURCES

We conducted our experiments on an Intel(R) Xeon(R) Platinum 8171M CPU @ 2.60GHz processor based system. The system consists of 2 processors, each with 26 cores running at 2.60GHz (52 cores in total) with 32KB of L1, 1024 KB of L2, 40MB of unified L3 cache, and 250 GB of memory. Besides, we use a single Nvidia RTX3090 GPU to facilitate the training procedure. The operating system is Ubuntu 16.04. Totally, we conduct experiments on 2500 seeds in 15 downstream tasks.

## E ADDITIONAL ANALYSIS OF ABLATION

The results of the complete ablation experiment for each environment are shown in Fig. 10. For the generalization ability of experiments, the results are based on the average performance of three types of exploration backbone with 2M steps of pre-training and 100k steps of fine-tuning (A total of 720 runs = 3 algorithms × 12 tasks × 5 seeds × 4 settings).

Overall, our default settings, i.e. reusing all components of the world models (encoder, dynamics model, reward model) and policies (actor and critic) works best. The most important modules in EUCLID are dynamics models and actor as these two always get a performance gain in all tasks, whereas reusing critic and reward predictor can improve performance only in some domains.

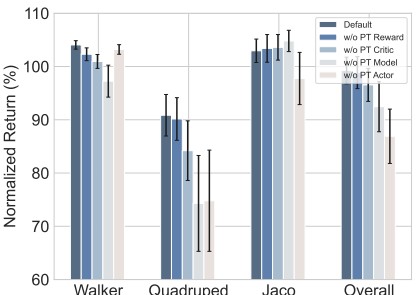

Figure 10: **Ablation.** Comparison with agents based on different pre-trained components for fine-tuning. The results are based on the average performance of three types of exploration backbone with 2M steps of pre-training.

Using pre-trained policies to initialize downstream policies shows significant performance in Quadruped and Jaco domains, which illustrates the synergistic effect of model based planning and pre-trained policy. Exploration policies trained by intrinsic rewards during the pre-training phase usually exhibit behaviours that are beneficial for downstream tasks, e.g. agents in the Quadruped

Table 4: Hyper-parameters of the world model, policy, and planner.

| World model | Value |
|---|---|
| Batch size | 1024 |
| Max buffer size | 1e6 |
| Latent dim | 50 (default) |
| | 100 (Humanoid) |
| MLP hidden dim | 256 (Encoder) |
| | 1024 (otherwise) |
| MLP activation | ELU |
| Optimizer ($\theta$) | Adam |
| Learning rate | 1e-4 (PT) |
| | 1e-3 (FT) |
| Reward loss coefficient ($c_1$) | 0.5 |
| Consistency loss coefficient ($c_2$) | 2 |
| Value loss coefficient ($c_3$) | 0.1 |
| $\theta^-$ update frequency | 2 |
| Policy | Value |
| Seed steps | 0 (PT) |
| | 4000 (FT) |
| Discount factor ($\gamma$) | 0.99 |
| Action repeat | 2 (default) |
| | 4 (Quadruped) |
| Planning (Only for FT phase) | Value |
| Iteration | 6 |
| Planning horizon ($L$) | 5 |
| CEM population size | 512 |
| CEM elite fraction | 12 |
| CEM policy fraction (Policy/CEM) | 0.05 |
| CEM Temperature | 0.5 |

Table 5: hyper-parameters of multi-choice learning mechanism

| MCL hyper-parameter | Value |
|---|---|
| Num of prediction heads ($H$) | 4 |
| Regularization strength ($\alpha$) | 0.1 |
| specific interval time steps $\mathcal{T}$ | 500k |

domain trained by the Disagreement can exhibit behaviours such as tumbling and wobbly standing, while randomly initialised agents only fall and shake. Therefore, using such policies to guide the planning process during the fine-tuning phase can help the agents reduce exploration cost and increase sampling efficiency which also demonstrates the benefits of a hybrid strategy.

Since the state transition of the environment are shared during the pre-training and fine-tuning phases, it makes sense that pre-trained dynamics model would be more accurate than a model initialized randomly. Moreover, we empirically find that reusing both the critic and reward predictor at the beginning of FT phase can improve overall performance, although the intrinsic reward which reward predictor predicts in pre-training phase is quite different from the expected extrinsic return given by downstream tasks. Thus, we still maintain the reuse of critic and reward predictor by default and leave whether the pre-training of critic and reward predictor can benefit for downstream tasks with a theoretical guarantee as an open question.

## F    PSEUDO CODES OF EUCLID

We show the full process of EUCLID, including both pre-training and fine-tuning phases, in Algorithm 1.

---

**Algorithm 1:** Efficient Unsupervised Reinforcement Learning Framework with Multi-choice Dynamics Model (EUCLID)

---

1  **Input:** specific interval time steps $\mathcal{T}$, prediction head size $H$, regularization strength $\alpha$, current policy size $h = 0$, ensemble policy set $S = \emptyset$.

2  **Require:** Initialize all networks: Encoder $E_\theta$, Multi-headed dynamics $D_\theta$, Reward predictor $R_\theta$, Critic $Q_\theta$, Actor $\pi_\phi$, Replay buffer $\mathcal{D}$.

3  **Require:** Environment (env), $M$ downstream tasks $T_k$, $k \in [1, \ldots, M]$.

4  **Require:** Intrinsic reward $r^{\text{int}}$, extrinsic reward $r^{\text{ext}}$.

5  **Require:** pre-train $N_{\text{PT}} = 2M$ and fine-tune $N_{\text{FT}} = 100K$ steps.

6  *# Part 1 : Unsupervised Pre-training*

7  **for** $t = 0, 1, ..N_{PT}$ **do**

8       **if** $t == \mathcal{T} * h$ **then**

9           Initialize $\pi_\phi^h$ with weights of $\pi_\phi$, $h \leftarrow h + 1$

10          Extend ensemble policy set $S \leftarrow S \cup \pi_\phi^h$

11          Update average policy distribution $\widetilde{\pi}_\phi(\mathbf{z}_t) = \sum_{i=1}^h \pi_\phi^i(\mathbf{z}_t)/h$

12      Encoder state $\mathbf{z}_t = E_\theta(\mathbf{s}_t)$ and sample action $\mathbf{a}_t \sim \pi_\theta(\mathbf{z}_t)$

13      Apply action to the environment $\mathbf{s}_{t+1} \sim P(\cdot \mid \mathbf{s}_t, \mathbf{a}_t)$

14      Add transition to replay buffer $\mathcal{D} \leftarrow \mathcal{D} \cup (\mathbf{s}_t, \mathbf{a}_t, \mathbf{s}_{t+1})$

15      Sample a minibatch from replay buffer $\mathcal{D}$, compute intrinsic reward $r^{\text{int}}$ with exploration backbone

16      Update encoder, reward predictor, critic and dynamics model parameters $\theta$ with prediction head $h$                              $\triangleright$ see Eq. 3

17      Update actor $\phi$ by RL loss with additional diversity encouraging term          $\triangleright$ see Eq. 5

18 *# Part 2 : Supervised Fine-tuning*

19 **for** $T_k \in [T_1, \ldots, T_M]$ **do**

20      Initialize all networks with weights from the pre-training phase and an empty replay buffer $\mathcal{D}$

21      Select the most appropriate prediction head $h^*$ with the highest zero-shot reward

22      Fix head $h^*$ and corresponding policy $\pi_\phi^{h^*}$

23      Initialize fine-tuning actor $\pi_\phi^{\text{FT}}$ with weights of $\pi_\phi^{h^*}$

24      **for** $t = 1..N_{FT}$ **do**

25          Encoder state $\mathbf{z}_t = E_\theta(\mathbf{s}_t)$ and select action through planning $\mathbf{a}_t = Plan(\mathbf{z}_t)$ guided by $\pi_\phi^{\text{FT}}$

26          Apply action to the environment $\mathbf{s}_{t+1}, r_t^{\text{ext}} \sim P(\cdot \mid \mathbf{s}_t, \mathbf{a}_t)$

27          Add transition to replay buffer $\mathcal{D} \leftarrow \mathcal{D} \cup (\mathbf{s}_t, \mathbf{a}_t, r_t^{\text{ext}}, \mathbf{s}_{t+1})$

28          Update encoder, reward predictor, critic and dynamics model parameters $\theta$ with fixed prediction head $h^*$                              $\triangleright$ see Eq. 3

29          Update actor $\phi$ only by RL loss          $\triangleright$ see Eq. 10

30      Evaluate performance of RL agent on task $T_k$

---

## G  DETAILS OF MULTI-CHOICE LEARNING

To stabilize the training process, we fixed the equal number of ensemble policies and prediction heads of the multi-headed dynamics model as hyper-parameters in advance, and corresponded the policies to the prediction heads one by one. Our core intuition is that diverse policies yield diverse data distributions, and we use the different data distributions to train the model primitives separately to obtain a set of sub-optimal dynamics models with their specified region. In the beginning of the PT stage, we initialize an empty set of policies $S = \emptyset$ and a dynamics model $D_\theta$ with $H$ prediction head. We extend ensemble policy set $S \leftarrow S \cup \pi_\phi^h$, $h \in 1..H$ at every specific interval time steps $\mathcal{T}$ for total 2M pre-training steps where network parameters of $\pi_\phi^h$ is initialized by the weights of current $\pi_\phi$.

To allow ensemble policies to be as diverse as possible, we calculate the current average policy distribution $\widetilde{\pi}_\phi$ each time we add a member to the ensemble and design policy diversity encouraging terms $D_{\text{KL}}(\widetilde{\pi}_\phi(\mathbf{z}_t) \| \pi_\phi(\mathbf{z}_t))$ to encourage diversity of new policy. We balance each policy

of ensemble between the exploration and the diversity objectives that each model primitive models different skill space. After a complete PT phase, we are able to obtain the multi-headed dynamics model and the corresponding set of policies $[\pi_\phi^1(\mathbf{z}), \cdots, \pi_\phi^h(\mathbf{z})]$.

In the beginning of FT phase, we should select one head that can benefit the downstream learning the most. We follow the same procedure for competence-based method adaptation as in URLB (Laskin et al., 2021). During the first 4k steps, the zero-shot evaluation that we use is to interact with the environment for a whole episode by each prediction head and the corresponding pre-trained policy, get the episode extrinsic reward and select the prediction head with the largest reward as the expert. After this, we fix the prediction head $h^*$ and finetune world model parameters for the remaining steps.

## H  DETAILS OF EUCLID MIXTURE PLANNING

Following TDMPC (Hansen et al., 2022), we leverage imaginary trajectories both for planning and policy gradient. As for planning, we perform Model Predictive Control (Bemporad & Morari, 1999) method MPPI (Williams et al., 2015) to update parameters for a family of distributions using an importance weighted average of the estimated top-k sampled trajectories of expected return. Specifically, we use the cross entropy method (CEM) (Rubinstein, 1997) to optimize action sequences by iteratively re-sampling action sequences near the best performing sequences from the last iteration. It is worth noting that we plan at each decision step $t$ and execute only the first action. In addition, long-term model planning is computational costly and inaccurate, and model errors accumulate with horizon length (Lai et al., 2020). Therefore, we use short-term reward estimates generated by the learned model and use the $Q$ value function for long-term return estimates via bootstrapping. At the same time, we hope that the pre-trained policy can guide the planning process and improve control performance. For this, we additionally mix the trajectory samples generated by policy with the planning trajectories to guarantee the optimization when model is not that accurate. The detail pseudo code is given in Algorithm 2.

---

**Algorithm 2:** Policy Guided Planning Algorithm

1 **Require:** Encoder $E_\theta$, Latent dynamics $D_\theta$, Reward predictor $R_\theta$, Critic $Q_\theta$, Actor $\pi_\phi$.
2 **Require:** Initial parameters for $\mathcal{N}(\mu^0, (\sigma^0)^2)$.
3 **Require:** Planning trajectories num $N$ and policy trajectories num $N_\pi$.
4 **Require:** Current state $s_t$, rollout horizon $L$, iteration num $J$, elite trajectories num $k$.
5 Encoder state $\mathbf{z}_t = E_\theta(\mathbf{s}_t)$ ▷ trajectory starting state
6 # Iterate J rounds starting from initial distribution $\mathcal{N}(\mu^0, (\sigma^0)^2)$
7 **for** *each iteration $j = 1, ...J$* **do**
8 $\quad$ Sample $N$ trajectories of length $L$ from $\mathcal{N}\left(\mu^{j-1}, \left(\sigma^{j-1}\right)^2\right)$
9 $\quad$ Sample $N_\pi$ trajectories of length $L$ using $D_\theta$ and $\pi_\phi$
10 $\quad$ Estimating the cumulative discounted rewards for all trajectories, the first $L$ steps are estimated using the reward predictor $R_\theta$ and thereafter using $Q_\theta$
11 $\quad$ Select top-$k$ elite trajectories based on cumulative rewards
12 $\quad$ Update $\mu, \sigma$ by top-$k$ sampled trajectories using MPPI (Williams et al., 2015)
13 **return** $\mathbf{a} \sim \mathcal{N}\left(\mu^J, \left(\sigma^J\right)^2\right)$

---

## I  ANALYSIS OF MODEL-BASED RL PERSPECTIVE

In the model-based RL community, how to collect data for model fitting is the general challenge. Most of the traditional model-based RL (Zhang et al., 2019; Hafner et al., 2019) approaches use task behavior policies to obtain data and train the world models. Some approaches enhance the breadth of the model by designing additional intrinsic reward-driven goals as auxiliary tasks for learning to obtain more diverse samples (Seo et al., 2021; Mazzaglia et al., 2022). However, these approaches still face the problem of poor transferability, which requires complete retraining when faced with similar downstream tasks. And we focus on how to improve the sample efficiency of model-based RL in the perspective of unsupervised RL and fit general world models for multiple downstream tasks through reasonable reward-free pre-training and fine-tuning paradigms. EUCLID stands for a PT and FT

perspective, with multiple choice learning and a corresponding policy diversity-encouraging term, hoping to train a set of submodels with separate expert regions and select the most suitable model for the downstream task. In this way, we can reuse the pre-trained model many times, adapting quickly using fewer samples, rather than learning from scratch.

# J    ADDITIONAL EXPERIMENTS

## J.1    ABLATION OF MULTI-CHOICE LEARNING IN HUMANOID DOMAIN

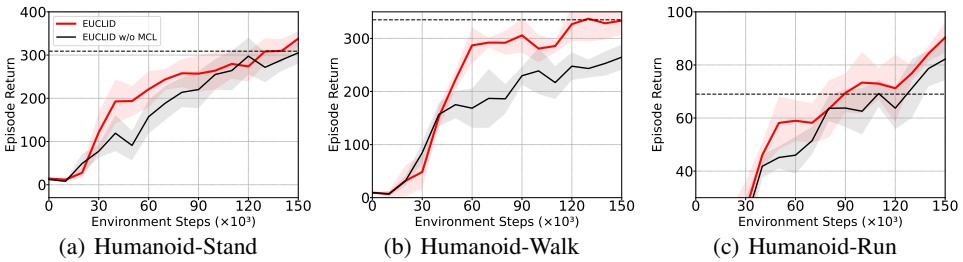

(a) Humanoid-Stand       (b) Humanoid-Walk       (c) Humanoid-Run

Figure 11: Learning curves of EUCLID and EUCLID w/o MCL on downstream tasks of the humanoid domain. Curves show the mean and 95% confidence intervals of performance across 5 independent seeds. The dashed reference lines are the asymptotic performance of the Disagreement algorithm with 2M PT steps and 2M FT steps.

As shown in Fig. 11, we ablate and evaluate the role of the MCL mechanism in the humanoid domain. Experiments show that the MCL mechanism can achieve a stable improvement in humanoid domain, which certifies the positive effect of the MCL mechanism in complex reward functions and complex environments.

## J.2    STATE PREDICTION ERROR REGRESSION ANALYSIS

To further verify the effectiveness of pre-training and MCL mechanism for dynamics model, we collected expert data for 30 episodes (30000 steps) of quadruped-run tasks. Then, we train a dynamics model on the pre-collected quadruped dataset. In order to exclude other factors, we removed the encoder module in this experiment, and the prediction loss is defined by $\|D_\theta(s_t, a_t) - s_{t+1}\|_2^2$. Fig. 12 show the prediction error curves for up to 10000 training iteration, including randomly initialized, pre-trained with and without multi-choice learning using EUCLID.

Firstly, we find that pre-trained dynamics models have smaller prediction errors than randomly initialization, especially better when using multi-choice learning. Secondly, the pre-trained dynamics models converge more quickly. Thirdly, When the number of training iterations is large enough, a single general pre-trained dynamics model (without MCL) and a randomly initialized dynamics model eventually converge to similar prediction error magnitude, while a multi-headed dynamics model using the MCL mechanism (selecting specific head) signifi-

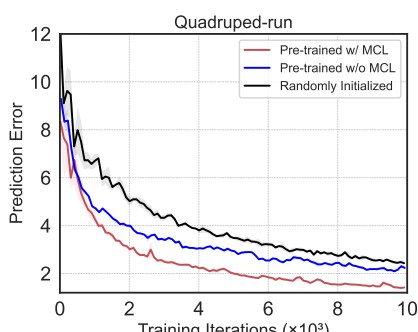

Figure 12: We collect expert data for quadruped-run tasks to train a dynamics model and report the prediction error curves on the held-out dataset. Curves show the mean and 95% confidence intervals of performance across 5 independent seeds.

cantly reduces the final prediction error. This indicates that EUCLID is able to predict the dynamics corresponding to the downstream task more accurately.

### J.3 EXTENSION EXPERIMENTS IN THE PIXEL-BASED URLB

To further demonstrate that EUCLID is a general and effective framework that can significantly accelerate the learning efficiency of downstream tasks through pre-training, we conducted additional pixel-based URLB experiments on four tasks. As for baseline in model-free manner, we opt for DrQ-v2 (Yarats et al., 2021b) (which can be simply viewed as DDPG + Image Augmentation for image inputs) as our base optimization algorithm to learn from images. In the pre-training phase, we consistently use disagreement as the backbone of exploration. The results shown in Table 6 are means over 3 seeds. In the table, TDMPC@100k and DrQ-v2@100k represent agents with 100k training steps and without pre-training. DrQ-v2 (Disagreement) and EUCLID (Disagreement) represent agents after 500k reward-free pre-training and 100k fine-tuning. The results show that EUCLID can also lead significant performance improvement in visual control tasks.

Table 6: Comparisons on the URLB with image inputs. Results for DrQ-v2@100k and DrQ-v2 (Disagreement) are obtained from (Laskin et al., 2021).

| Task | DrQ-v2@100k | DrQ-v2 (Disagreement) | TDMPC@100k | EUCLID (Disagreement) |
|---|---|---|---|---|
| Walker-Flip | 81±23 | 360±16 | 403±30 | **622±37** |
| Walker-Run | 41±11 | 131±19 | 201±15 | **323±41** |
| Walker-Stand | 212±28 | 398±65 | 793±40 | **950±15** |
| Walker-Walk | 141±53 | 348±46 | 573±43 | **743±38** |

