# OpenReview forum: "EUCLID: Towards Efficient Unsupervised Reinforcement Learning with Multi-choice Dynamics Model"
_ICLR.cc/2023/Conference — ICLR 2023 poster_

### Official Review · Reviewer_wBqh · 2022-10-23

**Confidence:** 3
**Correctness:** 3
**Technical Novelty And Significance:** 3
**Empirical Novelty And Significance:** 3
**Recommendation:** 6

**Clarity, Quality, Novelty And Reproducibility:**

Clarity:
- A few technical sections could have been presented more clearly. For example, particularly I think it could be more clear how the policy that is described in Section 3.1 is used (my understanding is that it is both an exploration policy during pretraining and also used to generate candidate action samples during finetuning).
- The context for Figure 1 is confusing – it’s not clear which method was used for pretraining in this example.
- Another point is that the statement in the introduction is that “pre-training via diverse exploration is not enough for guaranteeing to facilitate downstream learning”. But doesn’t this method also fall into the category of pre-training using diverse exploration?

Quality: The experimental evaluations seem to have been thoroughly executed and there are many ablation analyses, which I find helpful. See my points in “Weaknesses”.

Novelty: I think the novelty of the work is somewhat limited when compared to TDMPC. The combination of a TDMPC-like method with unsupervised RL algorithms is novel to me. The other main contribution claimed is the multi-choice learning framework, but as mentioned earlier, I am not convinced of its effectiveness.

The paper describes the implementation in significant detail and appears reproducible.

**Strength And Weaknesses:**

Strengths:
- The idea behind the method of the paper is well-motivated, and it conceptually makes sense that a pre-trained dynamics model would be one way to take advantage of an unsupervised pre-training phase.
- The EUCLID method is agnostic to the choice of intrinsic reward, as the authors point out, and the empirical study on the choice of unsupervised RL method is quite comprehensive.
Weaknesses:
- I think that the paper in its current form overclaims the contributions of the multiple-choice dynamics model learning for the final performance. In Table 1, it appears that EUCLID without MCL achieves very similar performance (often within the error bars) as EUCLID with MCL. Although the qualitative results presented show that certain task heads perform better than the other ones, it seems reasonable for that to happen because the heads are specifically trained to be better at specific tasks. But the claim that “This indicates that multi-choice learning reduces the prediction error of the downstream tasks corresponding to local dynamics and obtains better results” doesn’t seem substantiated.
- I also think that the results for TDMPC @ 100k, which are in the appendix, should be included in Table 1. Because EUCLID uses many of the components of TDMPC, the performance of TDMPC @ 100k is actually quite competitive with EUCLID.
- The method is only demonstrated on tasks from state, although because of the world model formulated, it seems like it would be readily transferable to learning from pixels. I think it would be helpful to see results on image-based tasks to show the methods’ efficacy.

**Summary Of The Paper:**

This paper presents a method called EUCLID for the unsupervised reinforcement learning (URL) setting, that consists of two phases: an unsupervised pretraining phase, and then a downstream task where task-specific rewards are provided. The method uses a model-based pretraining approach, where the model has multiple heads that are trained to be experts on particular subsets of the state space. During finetuning, the model is used in a model-based planning setup to complete downstream tasks. The method is evaluated on state-based tasks from the URLB benchmark. The paper also contributes empirical studies of which unsupervised RL benchmarks improve the performance of EUCLID and ablation studies of the method.

**Summary Of The Review:**

In summary, I think this is a nice paradigm for the unsupervised RL setting and includes thorough experimental evaluation and ablations. However, I have a few concerns regarding the novelty and improvements compared to TDMPC, effectiveness of the multi-headed model approach, and clarity of the paper. Therefore I think the paper currently shouldn’t be accepted, but am willing to change my score based on discussion.

---

> ### Author Response · Authors · 2022-11-14
> **Response to Reviewer wBqh (Part 3/3)**
>
> **[Q3: It would be helpful to see results on image-based tasks to show the methods’ efficacy.]**
>
> We agree with the reviewer that the addition of an image based experimental environment can increase the generality of the EUCLID method, and we have added the results of image-based URLB in the revision **(see Table.6, page 21, revised version paper)**.
>
> We conducted additional pixel-based URLB experiments on four tasks. As for baseline in model-free manner, we opt for DrQ-v2 (Yarats et al, 2021) (which can be simply viewed as DDPG + Image Augmentation for image inputs) as our base optimization algorithm to learn from images. In the pre-training phase, we consistently use disagreement as the backbone of exploration. The results are shown in the table below. In the table, TDMPC@100k and DrQ-v2@100k represent agents with 100k training steps and without pre-training. DrQ-v2 (Disagreement) and EUCLID (Disagreement) represent agents after 500k reward-free pre-training and 100k fine-tuning. The results show that EUCLID can also lead significant performance improvement in visual control tasks.
>
> |   **Task**   	| **DrQ-v2@100k** 	| **DrQ-v2 (Disagreement)** 	| **TDMPC@100k** 	| **EUCLID (Disagreement)** 	|
> |  ----  | ----  | ----  | ----   | ----  |
> |  Walker-Flip 	|      81±23      	|           360±16          	|     403±30     	|         **622±37**        	|
> |  Walker-Run  	|      41±11      	|           131±19          	|     201±15     	|         **323±41**        	|
> | Walker-Stand 	|      212±28     	|           398±65          	|     793±40     	|         **950±15**        	|
> |  Walker-Walk 	|      141±53     	|           348±46          	|     573±43     	|         **843±38**        	|
>
>
> **[Q4: The results for TDMPC@100k should be included in Table.1]**
>
> Thanks for your constructive suggestion. We previously placed the comparison of TDMPC@100k results in **Fig. 4 (Grey bar)**, considered as a comparison experiment between MBRL and URL baseline. We agree with the reviewers' comments and we have added them to the revised version **(See Table.1, page 8, revised version paper)**.
>
> **[Q5: How to use the policy which is described in Section 3.1? My understanding is that it is both an exploration policy during pretraining and also used to generate candidate action samples during finetuning]**
>
> Yes, you are absolutely correct in your understanding! In section 3.1, we focus on introducing the model structure (both the world model and the policy model). We put how the PT phase and the FT phase use the policy to interact with the environment in section 3.2 for description, and provide more details about hybrid planning in Appendix H.
>
> **[Q6: It’s not clear which method was used for pre-training in Figure.1]**
>
> Thanks to the reviewer for pointing out this misleading point, we have re-uploaded this figure in revision **(See Fig.1, page 2, revised version paper)**. The blue line represents EUCLID (Disagreement), while the orange line represents pre-training with Disagreement.
>
> **[Q7: About the statement “pre-training via diverse exploration is not
> enough for guaranteeing to facilitate downstream learning”]**
>
> Sorry for the confusion. In the introduction, we evaluate the previous model free URL approach and show potential limitations in existing URL approaches: ***pre-training via diverse exploration is not enough for guaranteeing to facilitate downstream learning.***
>
> We want to illustrate that **only** diverse exploration **for pre-training policy** is **not enough** to guarantee the performance improvement of downstream tasks. Because only diverse exploration is not sufficient, we propose EUCLID: **EUCLID = diverse exploration + model fused paradigm + MCL**.
>
> To avoid misunderstanding, we modify it in the revision as follows **(See red text in the updated manuscript, page 1)**:
> >Only pre-training policy via diverse exploration is not enough to guarantee to facilitate downstream learning.
>
> We hope this can be more clear for the readers to follow.
>
> ---
> Reference:
>
> [1] Barth-Maron G, et al. Distributed distributional deterministic policy gradients. ICLR2018.
> [2] Hafner D, et al. Dream to control: Learning behaviors by latent imagination. ICLR2020.
> [3] Hansen N, et al. Temporal difference learning for model predictive control. ICML2022.
> [4] Laskin M, et al. URLB: Unsupervised reinforcement learning benchmark. NeurIPS2021 Dataset Track.
> [5] Yarats D, et al. Mastering visual continuous control: Improved data-augmented reinforcement learning. ICLR2022.

---

> > ### Comment · Reviewer_wBqh · 2022-12-06
> > **Response to authors**
> >
> > Thank you for your detailed responses, clarifications, and efforts to improve the paper. Given the clarifications for the experimental results and especially additional pixel-based results, many of my concerns have been addressed. However, I am still not entirely convinced of the role of MCL as the comparison in Appendix J.2 does not seem to account for other factors such as model capacity, as brought up by yoch; so I feel that the importance of this component is overstated in the current version of the text.
> >
> > I have updated my score accordingly.

---

> > > ### Author Response · Authors · 2022-12-07
> > > **Sincererly thank you for your recognition**
> > >
> > > We sincererly thank you for your constructive suggestions and efforts. The quality of our paper has improved a lot with your help. Next, we will still improve the quality of the paper based on your suggestions. Many thanks for your kind suggestions and warm help.

---

> ### Author Response · Authors · 2022-11-14
> **Response to Reviewer wBqh (Part 2/3)**
>
> **[Q2: Novelty and improvements compared to TDMPC]**
>
> **Novelty**: We would like to clarify that our novelty is based on the following five points:
>
> - To the best of our knowledge, the role of model-based RL in unsupervised RL setting was not obvious. In particular, we believe that our **effective and simple** MBRL technique can be an important milestone in the URL community, as model fitting can be easily performed using the huge amount of unlabeled data in the pre-training stage **with no additional cost**, and the related work on URL in model free manner is **completely orthogonal and easily combined**.
> - EUCLID is a generic model-fused unsupervised reinforcement learning framework with MCL whose **world model components can be easily replaced by other model structures such as dreamer**  (Hafner et al, 2019). EUCLID lies in general unsupervised pre-training and fine-tuning to quickly adapt to multiple downstream tasks, while TDMPC (Hansen et al, 2022) focuses on training a world model for a specific task.
> - EUCLID **alleviates the oscillation in performance through longer PT steps (mismatch issue)** while prior URL works failed, while (Laskin et al, 2021) see this as potentially the **biggest drawback** of current unsupervised RL approaches.
> - Furthermore, We further introduce multiple choice learning, **including multi-headed dynamics model, diversity encouraging term for unsupervised exploration and zero-shot selection method**. The experimental results show that each prediction head indeed covers the different expert regions (Fig.7 (a)) and has more accurate prediction performance in the expert regions (response of Q1).
> - Although the world model structure of EUCLID uses the existing components, it is important that the work **investigate the influence of different mainstream URL designs** for world models and analyze some design choice about **how each part of the world model affects the ultimate performance** in the pre-training and fine-tuning procedure. This can all have a significant impact on the final performance.
>
> **Improvements**: We observe that in simple domains (walker), both TDMPC@100k and EUCLID can reach near-maximal performance. However, under complex dynamics domain such as quadruped and humanoid, and sparse reward tasks such as jaco, the EUCLID framework is able to **achieve stable and significant performance gains**, and we present the average performance gains (average normalized score) for each domain in the following table:
>
> | **Domain** 	| **TDMPC@100k (Normalized score)** 	| **EUCLID (Normalized score)** 	| **Improvement** 	|
> |  ----  | ----  |  ----  | ----  |
> | Walker     	| 101.4±1.6%                        	| **104.6±0.4%**                	| ↑3.2%           	|
> | Quadruped  	| 74.7±9.8%                         	| **97.6±1.1%**                 	| ↑22.9%          	|
> | Jaco       	| 86.9±6.4%                         	| **109.7±2.0%**                	| ↑22.8%          	|
>
> | **Domain** 	| **TDMPC@150k (total reward)** 	| **EUCLID (total reward)** 	| **Improvement** 	|
> |  ----  | ----  |  ----  | ----  |
> | Humanoid   	| 173±93                       	| **761±47**               	| ↑339.9%         	|
>
>
> It is worth noting that EUCLID can be particularly rewarding in a humanoid environment (**339.9% improvement**), as naive TDMPC@150k struggles to achieve competitive performance.

---

> ### Author Response · Authors · 2022-11-14
> **Response to Reviewer wBqh (Part 1/3)**
>
> We thank the reviewer for the insightful and useful feedback, please see the following for our response.
>
> **[Q1: The paper overclaims the contributions of the multiple-choice dynamics model learning for the final performance.]**
>
> The reviewer mentioned that "In Table 1, it appears that EUCLID without MCL achieves very similar performance (often within the error bars) as EUCLID with MCL."
>
> For this, we believe the reviewer is concerning about **the performance gains incurred by the MCL** in Table.1. On one hand, in easy task domains (e.g., walker), EUCLID w/o MCL can reach near-maximal performance, and that is why the advantage of MCL seems insignificant.
> An evidence is that, in the prior work (Barth-Maron et al, 2018), D4PG is trained with 1e9 steps (much longer than our setting i.e., 1e5) and scores 946 in walker-walk (see more in the following table).
>
> | **Task**     	| **D4PG(1e9)** 	| **EUCLID w/o MCL(1e5)** 	| **EUCLID(1e5)** 	|
> |  ----  | ----  | ----  | ----  |
> | walker-walk  	| 946           	| 967±4                   	| 972±1           	|
> | walker-run   	| 841           	| 765±10                  	| 770±9           	|
> | walker-stand 	| 976           	| 985±1                   	| 985±1           	|
>
>
> On the other hand, we hypothesize that there are usually more shared dynamics in relatively easy tasks, which leads to a small performance difference (and equally good) per prediction head, and we empirically **find more complex task dynamics (harder to predict) where the MCL mechanism performs better**, such as the quadruped in Table.1:
>
> | **Task**       	| **EUCLID w/o MCL(1e5)** 	| **EUCLID(1e5)** 	| **Improvement** 	|
> |  ----  | ----  |  ----  | ----  |
> | Quadruped-run  	| 651±35                  	| **735±16**      	| ↑13%            	|
> | Quadruped-walk 	| 874±42                  	| **925±6**       	| ↑6%             	|
>
> In addition, we provide additional experiments of performance **ablation of MCL on the humanoid domain** (**super hard**). We provide experimental details and reward curves in **Appendix J.1 (page 20, revised version paper)**, and briefly present the results here:
>
> | **Task** | **EUCLID w/o MCL** 	| **EUCLID** 	| **Improvement** 	|
> |  ----  | ----  |  ----  | ----  |
> | Humanoid-stand | 305±29             	| **338±19** 	| ↑11%            	|
> | Humanoid-walk  |  264±27             	| **333±22** 	| ↑26%            	|
> | Humanoid-run  |  82±9               	| **90±6**   	| ↑10%            	|
>
> We observe that in the humanoid domain, the use of MCL results in higher performance.
>
> ==========================================================
>
> Another reviewer's comment is that: " 'This indicates that multi-choice learning reduces the prediction error of the downstream tasks corresponding to local dynamics and obtains better results' doesn’t seem substantiated."
>
> Thanks for pointing this out, more analysis of the model error experiments are added in the revision. We investigated the effect of MCL and found that the MCL mechanism (including multi-headed dynamics model and diversity-encouraging loss) can further **reduce the prediction error** of the downstream tasks.
>
> In this experiment, we collect expert data for the quadruped-run task and train a regression model predicting the next state. **We report the prediction error on held-out datasets, including Randomly Initialized, Pre-trained with and without MCL. See Appendix J.2 (page 20, revised version paper)** for detailed experimental results and setup.
>
> | **Training Iterations** 	| **0**    	| **10000** 	|
> |  ----  | ----  |  ----  |
> | Randomly Initialized    	| 12.23    	| 2.43      	|
> | Pre-trained w/o MCL     	| 9.29     	| 2.24      	|
> | **Pre-trained w/ MCL**  	| **8.34** 	| **1.42**  	|
>
> We have the following three findings:
>
> - **Firstly**, we find that pre-trained dynamics models have smaller prediction errors than randomly initialization at the very beginning (iteration 0), especially better when using MCL.
>
> - **Secondly**, the pre-trained model converges more quickly **(see Fig.12, page 20, revised version paper)**.
>
> - **Thirdly**, when the number of training iterations is large enough (iteration 10000), a single general pre-trained dynamics model and a randomly initialized dynamics model eventually converge to similar prediction error magnitude, while a dynamics model using the MCL mechanism (selecting specific head) significantly reduces the final prediction error.
>
> This shows that MCL can additionally reduce the prediction error of downstream tasks, indirectly leading to the fast adaptation of the model.

---

> ### Author Response · Authors · 2022-11-26
> **Resolving any pending concerns**
>
> We would really aporeciate it if the reviwers can let us know whether thev have any further concerns and whether our response addressed some/aof their concerns, We will try to address them before the discussion period ends, Thanks!

---

### Official Review · Reviewer_E7J9 · 2022-10-24

**Confidence:** 4
**Correctness:** 3
**Technical Novelty And Significance:** 3
**Empirical Novelty And Significance:** 3
**Recommendation:** 6

**Clarity, Quality, Novelty And Reproducibility:**

Clarity:

The presentation is clear and easy to understand.

Novelty:
The idea is new in this area although it has been proposed in dynamics generalization task.

Reproducibility:

The authors provide enough details to reproduce their methods.

**Strength And Weaknesses:**

Strength:

1. The motivation is clear and the proposed method make senses.
2. The idea is novel in this field, although it has been proposed in dynamics generalization task.
3. The experimentally comparisons are comprehensive and the improvement is impressive.

Weakness:

1.  This paper employs a multi-choice model to learn the environmental dynamics; however, Table 1 demonstrates that the performance difference between with MCL and without MCL is marginal, which raises my concern that MCL is unnecessary for this paradigm. If so, could "multi-choice" be removed from the title? Can authors provide further clarification?
2. According to my knowledge, the scale of rewards in the training phase and the fine-tune phase are vastly different. I am concerned that the scale difference may be greater than the differences between prediction heads in some situations, rendering the MCL ineffective. Can authors provide additional experiment information and observations?

**Summary Of The Paper:**

This paper introduces EUCLID, a new paradigm for the unsupervised reinforcement learning task. In the pre-training phase, the paradigm first introduces a multi-choice prediction model to learn the dynamics of the environment. The learned dynamics can adapt the policy to downstream tasks efficiently and effectively. The experimental results show significant improvements over baselines.

**Summary Of The Review:**

Overall, I think that the paper is above the accpetance bar considering the idea, difficulty to impletement and the experimental improvements,  but I still have some concerns about the metods details.

---

> ### Author Response · Authors · 2022-11-14
> **Response to Reviewer E7J9 (Part 2/2)**
>
>
> **[Q2: Concerns about MCL when the scale of rewards in PT and FT phase are vastly different]**
>
> Due to the special features of the unsupervised pre-training setup, the pre-training phase is reward agnostic, so the reward function for the downstream task will be vastly different from the intrinsic reward function used for pre-training. This is exactly what URL hopes to study (Laskin et al, 2021), by exploring a lot of diversity in the pre-training phase and adapting quickly in the new reward function task. When the reward function of the downstream task is complex (quadruped, humanoid) or sparse (jaco), the MCL mechanism can still work (see Q1 of response).
>
> **[Q3: The idea is novel in this field, although it has been proposed in dynamics generalization task.]**
>
> While prior dynamics generalization works related to Model-based RL have focused on meta-learning setting, e.g. CaDM (Lee et al, 2020), T-MCL (Seo et al, 2020), DOMINO (Mu et al, 2022). **Our study extends it to the unsupervised pre-training paradigm for URL and alleviates the oscillation in performance with longer pre-training steps.** Unlike T-MCL, we not only use a multi-headed dynamics model, but also design a novel diversity-encouraging term for diverse exploration and give more analysis (shared backbone vs. multiple models), allowing different prediction heads to cover as much different expert regions as possible, and our supplementary experiments (**See Appendix J.2, page 20, revised version paper**) demonstrate that the MCL mechanism can indeed reduce prediction errors.
>
>
> ---
>
> Reference:
>
> [1] Barth-Maron G, et al. Distributed distributional deterministic policy gradients. ICLR2018.
> [2] Lee K, et al. Context-aware Dynamics Model for Generalization in Model-Based Reinforcement Learning. ICML2020.
> [3] Seo Y, et al. Trajectory-wise multiple choice learning for dynamics generalization in reinforcement learning. NeurIPS2020.
> [4] Mu Y, et al. DOMINO: Decomposed Mutual Information Optimization for Generalized Context in Meta-Reinforcement Learning. NeurIPS2022.
> [5] Laskin M, et al. URLB: Unsupervised reinforcement learning benchmark. NeurIPS2021 Dataset Track.

---

> ### Author Response · Authors · 2022-11-14
> **Response to Reviewer E7J9 (Part 1/2)**
>
> We thank the reviewer for the recognition of our work, and please see the following for our response.
>
> **[Q1: Performance improvement of MCL in Table.1 is marginal]**
>
> We notice that the reviewer has expressed concerns about **the performance gains incurred by MCL**. On one hand, in easy task domains, such as walker, EUCLID w/o MCL is already able to reach near-optimal performance. We quote the convergence performance comparison of **D4PG (1e9)** (Barth-Maron et al, 2018) to illustrate this point.
>
> | **Task**     	| **D4PG (1e9)** 	| **EUCLID w/o MCL (1e5)** 	| **EUCLID (1e5)** 	|
> |  ----  | ----  |  ----  | ----  |
> | walker-walk  	| 946           	| 967±4                   	| 972±1           	|
> | walker-run   	| 841           	| 765±10                  	| 770±9           	|
> | walker-stand 	| 976           	| 985±1                   	| 985±1           	|
>
>
> On the other hand, there are usually more shared dynamics in relatively easy tasks, which leads to a small performance difference (and equally good) per prediction head, and **we empirically find more complex task dynamics (harder to predict) where the MCL mechanism performs better**, such as the quadruped in Table.1:
>
> | **Task**       	| **EUCLID w/o MCL(1e5)** 	| **EUCLID(1e5)** 	| **Improvement** 	|
> |  ----  | ----  | ----  | ----  |
> | Quadruped-run  	| 651±35                  	| **735±16**      	| ↑13%            	|
> | Quadruped-walk 	| 874±42                  	| **925±6**       	| ↑6%             	|
>
> In addition, we provide additional experiments of performance **ablation of MCL on the humanoid domain** (**super hard**). We provide experimental details and reward curves in **Appendix J.1 (page 20, revised version paper)**, and briefly present the results here:
>
> | **Task**       	| **EUCLID w/o MCL** 	| **EUCLID** 	| **Improvement** 	|
> |  ----  | ----  |  ----  | ----  |
> | Humanoid-stand 	| 305±29             	| **338±19** 	| ↑11%            	|
> | Humanoid-walk  	| 264±27             	| **333±22** 	| ↑26%            	|
> | Humanoid-run   	| 82±9               	| **90±6**   	| ↑10%            	|
>
> We observe that in the humanoid domain, the use of MCL results in greater performance gains.
>
> We also report the **additional prediction error experiments** on held-out datasets and find that the MCL mechanism can further **reduce the prediction error** of the downstream tasks. including Randomly Initialized, Pre-trained with and without MCL. See **Appendix J.2 (page 20, revised version paper)** of our revision for detailed experimental results and setup.
>
> | **Training Iterations** 	| **0**    	| **10000** 	|
> |  ----  | ----  |----  |
> | Randomly Initialized    	| 12.23    	| 2.43      	|
> | Pre-trained w/o MCL     	| 9.29     	| 2.24      	|
> | **Pre-trained w/ MCL**  	| **8.34** 	| **1.42**  	|
>
> We have the following three findings:
>
> - **Firstly**, we find that pre-trained dynamics models have smaller prediction errors than randomly initialization at the very beginning (iteration 0), especially better when using MCL.
>
> - **Secondly**, the pre-trained model converges more quickly **(see Fig.12, page 20, revised version paper)**.
>
> - **Thirdly**, when the number of training iterations is large enough (iteration 10000), a single general pre-trained dynamics model and a randomly initialized dynamics model eventually converge to similar prediction error magnitude, while a dynamics model using the MCL mechanism (selecting specific head) significantly reduces the final prediction error.
>
> In summary, MCL can improve sampling efficiency and reduce the prediction error of downstream tasks, which is a necessary component in the paradigm.

---

> ### Author Response · Authors · 2022-11-26
> **Resolving any pending concerns**
>
> We would really appreciate it if the reviewers can let us know whether they have any further concerns and whether our response addressed some/all of their concerns. We will try to address them before the discussion period ends. Thanks!

---

### Official Review · Reviewer_486X · 2022-10-25

**Confidence:** 4
**Clarity, Quality, Novelty And Reproducibility:** The paper is very clearly written. Th…
**Correctness:** 4
**Technical Novelty And Significance:** 2
**Empirical Novelty And Significance:** 3
**Recommendation:** 6

**Details Of Ethics Concerns:**

N/A.

**Strength And Weaknesses:**

Strength
* The pretraining and finetuning phases, with a specific design of multi-choice dynamics model, is well-motivated.
* The method achieves SOTA performance on standard benchmarks with high sample efficiency.
* Rationale behind several design choices, such as using multiple heads for dynamic model training, are empirically supported in experiments.

Weaknesses
* The method combines multiple components such as MBRL and MCL into a pretraining and finetuning framework. These components are directly taken from the existing literature. While the paper does offer comparisons of which backbones offer the best empirical results, the novelty of the work is limited.

**Summary Of The Paper:**

This work proposes a framework for joint pretraining of dynamic model and exploration policy and the adaptation to downstream tasks. The method achieves superior performance compared to prior works with a high sample efficiency.

**Summary Of The Review:**

This work proposes a model-based training framework that leverages dynamic information and skills learnt from the pretraining phase to speed up learning in downstream tasks. The novelty of the framework is limited since the components in this framework are directly taken from the literature, but I think this itself is not a big limitation given the good experimental performance in standard benchmarks.

---

> ### Author Response · Authors · 2022-11-14
> **Response to Reviewer 486X**
>
> We thank the reviewer for the recognition of our work, and please see the following for our response.
>
> **[Q1: The method combines multiple components such as MBRL and MCL into a pre-training and fine-tuning framework. the novelty of the work is limited.]**
>
> We would like to clarify that our novelty is based on the following five points:
>
> - To the best of our knowledge, the role of model-based RL in unsupervised RL setting was not obvious. In particular, we believe that our **effective and simple** MBRL technique can be an important milestone in the URL community, as model fitting can be easily performed using the huge amount of unlabeled data in the pre-training stage **with no additional cost**, and the related work on URL in model free manner is **completely orthogonal and easily combined**.
> - EUCLID is a generic model-fused unsupervised reinforcement learning framework with MCL whose **world model components can be easily replaced by other model structures such as dreamer**  (Hafner et al, 2019). EUCLID lies in general unsupervised pre-training and fine-tuning to quickly adapt to multiple downstream tasks, while TDMPC (Hansen et al, 2022) focuses on training a world model for a specific task.
> - EUCLID **alleviates the oscillation in performance through longer PT steps (mismatch issue)** while prior URL works failed, while (Laskin et al, 2021) see this as potentially the **biggest drawback** of current unsupervised RL approaches.
> - Multi-choice learning mechanism in EUCLID includes not only the multi-headed dynamics models similar to T-MCL (Seo et al, 2020), but also **diversity encouraging terms that are suitable for unsupervised exploration**. The experimental results show that each prediction head indeed covers the different experts and has more accurate prediction performance in the expert regions.
> - Although the world model structure of EUCLID uses the existing components, it is important that the work **investigate the influence of different mainstream URL designs** for world models and analyze some design choice about **how each part of the world model affects the ultimate performance** in the pre-training and fine-tuning procedure. This can all have a significant impact on the final performance.
>
> We hope this improves your opinion of our work. Thank you again for your reviewing efforts.
>
> ---
>
> Reference:
>
> [1] Hansen N, et al. Temporal difference learning for model predictive control. ICML2022.
> [2] Seo Y, et al. Trajectory-wise multiple choice learning for dynamics generalization in reinforcement learning. NeurIPS2020.
> [3] Hafner D, et al. Dream to control: Learning behaviors by latent imagination. ICLR2020.
> [4] Laskin M, et al. URLB: Unsupervised reinforcement learning benchmark. NeurIPS2021 Dataset Track.

---

> ### Author Response · Authors · 2022-11-26
> **Resolving any pending concerns**
>
> We would really appreciate it if the reviewers can let us know whether they have any further concerns and whether our response addressed some/all of their concerns. We will try to address them before the discussion period ends. Thanks!

---

### Official Review · Reviewer_yoch · 2022-11-04

**Confidence:** 4
**Correctness:** 3
**Technical Novelty And Significance:** 2
**Empirical Novelty And Significance:** 3
**Recommendation:** 6

**Clarity, Quality, Novelty And Reproducibility:**

The exposition is mostly clear, but the method relies on prior work as solutions to subproblems, details of which are not always fully explained; for example, there is not much explanation for how the correct expert is chosen from only zero-shot model evaluation (which seems to follow Seo et al 2020).

The novelty is limited in so far as the approach combines a variant of existing model-based approaches (such as TDMPC, Hansen et al 2021, and T-CML, Seo et al 2020) with existing exploration strategies.

It would be useful to verify that the positive effect of multiple choice is not simply due to increase in capacity of the dynamics model.


**Strength And Weaknesses:**

Strengths:

Experiments demonstrate that the resulting pretraining procedure is effective on URLB, compared to baselines. In particular, their approach requires significantly fewer samples under the downstream task.

Experiments demonstrate that unlike the model-free URL pretraining approaches they compare to, the resulting model benefits from more pre-training steps (and data) in a monotonic manner.

The authors provide a comparison between a significant number of exploration strategies (APT, Disagreement, DIAYN). While this is certainly an added bonus, this also introduces many hyper-parameter tuning challenges.

The authors provide additional comparisons on tasks based on humanoid, which are typically not considered in such pretraining settings.
In general, this work provides a useful case study of the model-based paradigm applied to the URLB benchmark.

Weaknesses:

The work is mainly positioned with respect to work in URL, but it also relates more broadly to literature in model-based RL, and the general challenge of collecting data for model fitting. It might be more impactful to consider prioritizing the model-based RL perspective and how the proposed method , in which case URLB may serve as one of many benchmarks.

The authors do not spend much time discussing how the mismatch problem can be circumvented with existing URL approaches, to be effectively transferred to downstream tasks; for example, skill discovery algorithms with discrete skill variables such as DIAYN (Eysenbach et al, 2018) can be interpreted as producing a set of experts which can then be treated as a mixture of experts by a master policy. Such an approach should serve as a baseline.

The novelty is limited in so far as the approach combines a variant of existing model-based approaches (such as T-CML, Seo et al 2020) with existing exploration strategies.


**Summary Of The Paper:**

This paper proposes an approach for pretraining an RL agent in a reward-free environment by fitting a dynamics model on the data generated by an exploration policy. The authors position this work in contrast to the body of prior work in unsupervised reinforcement learning (URL) involving model-free policy pretraining, which suffers from what they refer to as the “mismatch problem”: prior URL model-free approaches, like most exploration algorithms, optimize non-stationary reward functions and therefore end up oscillating in state distribution, meaning that transfer performance (which relies on overlap between the state distribution of the pretrained policy and that of the optimal policy for the target task) is highly dependent on the learning update at which the pretraining is stopped. The authors instead propose to pretrain a model-based policy by pretraining a dynamics model on data produced by an exploration policy; to improve the diversity of the pretraining data distribution for the dynamics model, the authors propose to use an ensemble of exploration policies which are additionally incentivized to visit different, and therefore specialize to, subspaces of the latent state space. The authors demonstrate that the pretrained learned dynamics model and policies can be transferred to downstream tasks by using the dynamics model for planning and selecting an expert from the ensemble for guiding the planner. In experiments on the URLB benchmark, they demonstrate that their method outperforms model-free URL baselines, and ablate choices pertaining to the exploration strategy and components of the objective for learning the dynamics model.

**Summary Of The Review:**

Overall, this work serves as an interesting case study of a model-based approach on the URLB benchmark. That said, the stated contribution of providing a novel model-fused approach for unsupervised pre-training is mainly positioned with respect to existing model-free approaches in the URL space, even though it ends up being quite related to existing work in model-based RL, and in particular, the challenge of obtaining data for fitting the dynamics model. Moreover, the comparison to transferring e.g. competence based URL methods by learning to control discovered skills is missing, which should serve as a stronger baseline.

---

> ### Author Response · Authors · 2022-11-14
> **Response to Reviewer yoch (Part 1/3)**
>
> We thank the reviewer for the insightful and useful feedback, please see the following for our response.
>
> **[Q1: How the mismatch problem can be circumvented with existing URL approaches?]**
>
> we would like to clarify that previous URL methods, including the skill discovery methods [DIAYN (Eysenbach et al, 2018), SMM (Lee et al, 2019), APS (Liu et al, 2021)], also face the *mismatch issue* (performance oscillation as the pre-training steps increase) in this kind of reward-free pre-training setting in some tasks and perform poorly in the URLB. (Laskin et al, 2021) conducted detailed experiments in URLB, and their results are shown in the following table (take jaco domain as an example, we refer to **Fig.4 and Appendix D of (Laskin et al, 2021)** for full results):
>
> | **Competence based method**  	| **100k** 	| **500k** 	| **2M**   	|
> |  ----  | ----  | ---- | ---- |
> | DIAYN (Eysenbach et al, 2018) 	| 0.26     	| 0.22     	| 0.11     	|
> | SMM (Lee et al, 2019)         	| 0.28     	| 0.33     	| 0.23     	|
> | APS (Liu et al, 2021)         	| 0.81     	| 0.63     	| 0.41     	|
> | **EUCLID (DIAYN) (ours)**      	| **0.87** 	| **0.91** 	| **0.94** 	|
>
> Thus, (Laskin et al, 2021) argue that this may be the biggest drawback of current unsupervised RL methods and to the best of our knowledge, there are no published experimental results reporting that they mitigate this problem in URL, including skill discovery methods. We train a more accurate dynamics model in the PT phase and then use it for planning decisions, instead of relying only on the unstable pre-trained policies, alleviating the mismatch problem, see results of Fig.6.
>
> **[Q2: the comparison to competence-based URL methods by learning to control discovered skills is missing, which should serve as a stronger baseline.]**
>
> We implement EUCLID with DIAYN as the exploration backbone in the paper and compare it with the original DIAYN, **see Fig. 4 and Fig. 6 for details.** When DIAYN is combined with EUCLID, it can benefit from longer pre-training steps to obtain better monotonic incremental results and can achieve high performance of 100k.
>
> We focus on the fact that although skill discovery methods learn different skill behaviors, their performance (both DIAYN and EUCLID(DIAYN)) still underperforms compared to knowledge-based methods. We hope to improve the exploration efficiency of competence-based algorithms in future work to obtain more meaningful skill behaviors corresponding to the world model and help to fast adaptation.
>
> **[Q3: It might be more impactful to consider prioritizing the model-based RL perspective.]**
>
> We agree with the reviewer that it might be more meaningful to further discuss our work from the perspective of model-based RL. From this perspective, our work has some overlap with the topical issues of model-based RL, and we add an extension discussion in the revision (**see Appendix I, page 19, revised version paper**):
>
> >In the model-based RL community, how to collect data for model fitting is a general challenge. Most of the traditional MBRL approaches use task behavior policies to obtain data and train the world models. Some approaches enhance the breadth of the model by designing additional intrinsic reward-driven goals as auxiliary tasks for learning to obtain more diverse samples (Seo et al, 2021; Mazzaglia et al, 2022). However, these approaches still face the problem of poor transferability, which requires complete retraining when faced with similar downstream tasks. And we focus on **how to improve the sample efficiency of model-based RL in the perspective of unsupervised RL and fit general world models for multiple downstream tasks through reasonable reward-free pre-training and fine-tuning paradigms.** EUCLID stands for a PT and FT perspective, with multiple choice learning and a corresponding policy diversity-encouraging term, hoping to train a set of submodels with separate expert regions and select the most suitable model for the downstream task.  In this way, we can reuse the pre-trained model many times, adapting quickly using fewer samples, rather than learning from scratch.

---

> ### Author Response · Authors · 2022-11-14
> **Response to Reviewer yoch (Part 2/3)**
>
> **[Q4: verify the positive effect of multiple choice]**
>
> We appreciate the reviewer for raising the suggestion and we provided additional experiments to make a better convey of the effect of multi-choice learning. We investigated the effect of MCL and found that the MCL mechanism (including multi-headed dynamics model and diversity-encouraging loss) can further **reduce the prediction error** of the downstream tasks, which may be an important reason why it can achieve good results.
>
> We collect expert data for the quadruped-run task and train a dynamics model predicting the next state. We report the prediction error on held-out datasets, including Randomly Initialized, Pre-trained with and without MCL. **see Appendix J.2 (page 20)** of our revision for detailed experimental setup and results.
>
> | **Training Iterations** 	| **0**    	| **10000** 	|
> |  ----  | ----  | ---- |
> | Randomly Initialized    	| 12.23    	| 2.43      	|
> | Pre-trained w/o MCL     	| 9.29     	| 2.24      	|
> | **Pre-trained w/ MCL**  	| **8.34** 	| **1.42**  	|
>
> We have the following three findings:
>
> - **Firstly**, we find that pre-trained dynamics models have smaller prediction errors than randomly initialization at the very beginning (iteration 0), especially better when using MCL.
>
> - **Secondly**, the pre-trained model converges more quickly **(see Fig.12, page 20, revised version paper)**.
>
> - **Thirdly**, when the number of training iterations is large enough (iteration 10000), a single general pre-trained dynamics model and a randomly initialized dynamics model eventually converge to similar prediction error magnitude, while a dynamics model using the MCL mechanism (selecting specific head) significantly reduces the final prediction error.
>
> This shows that MCL can additionally reduce the prediction error, indirectly leading to the fast adaptation of the model.
>
> **[Q5: The novelty is limited in so far as the approach combines a variant of existing model-based approaches with existing exploration strategies]**
>
> We would like to clarify that our novelty is based on the following five points:
>
> - To the best of our knowledge, the role of model-based RL in unsupervised RL setting was not obvious. In particular, we believe that our **effective and simple** MBRL technique can be an important milestone in the URL community, as model fitting can be easily performed using the huge amount of unlabeled data in the pre-training stage **with no additional cost**, and the related work on URL in model free manner is **completely orthogonal and easily combined**.
> - EUCLID is a generic model-fused unsupervised reinforcement learning framework with MCL whose **world model components can be easily replaced by other model structures such as dreamer**  (Hafner et al, 2019). EUCLID lies in general unsupervised pre-training and fine-tuning to quickly adapt to multiple downstream tasks, while TDMPC (Hansen et al, 2022) focuses on training a world model for a specific task.
> - EUCLID **alleviates the oscillation in performance through longer PT steps (mismatch issue)** while prior URL works failed, while (Laskin et al, 2021) see this as potentially the **biggest drawback** of current unsupervised RL approaches.
> - Multi-choice learning mechanism in EUCLID includes not only the multi-headed dynamics models similar to T-MCL, but also **diversity encouraging terms that are suitable for unsupervised exploration**. The experimental results show that each prediction head indeed covers the different expert regions and has more accurate prediction performance in the expert regions.
> - Although the world model structure of EUCLID uses the existing components, it is important that the work **investigate the influence of different mainstream URL designs** for world models and analyze some design choice about **how each part of the world model affects the ultimate performance** in the pre-training and fine-tuning procedure. This can all have a significant impact on the final performance.

---

> ### Author Response · Authors · 2022-11-14
> **Response to Reviewer yoch (Part 3/3)**
>
> **[Q6: There is not much explanation for how the correct expert is chosen from only zero-shot model evaluation]**
>
> We thank the reviewers for pointing out this lack of clarity, more explanations are added in the **Appendix G (red text, page 19, revised version paper)** and outlined as follows:
>
> >The zero-shot evaluation that we use is to interact with the environment for a whole episode by each prediction head and the corresponding pre-trained policy during the first 4k steps, get the episode reward and select the submodel with the largest reward as the expert for the remaining steps. And it is worth noting that because of the specificity of unsupervised pre-training, we cannot train a set of models strictly matching one-to-one with the downstream task (without a designed reward function). Therefore, our core goal is to find the prediction head that is closest to the downstream task.
>
> We hope this improves your opinion of our work. Thank you again for your reviewing efforts.
>
> ---
>
> Reference:
>
> [1] Eysenbach B, et al. Diversity is all you need: Learning skills without a reward function. ICLR2019.
> [2] Lee L, et al. Efficient exploration via state marginal matching. CoRR, abs/1906.05274, 2019.
> [3] Liu, Hao and Abbeel, Pieter. APS: active pretraining with successor features. ICML2021.
> [4] Laskin M, et al. URLB: Unsupervised reinforcement learning benchmark. NeurIPS2021 Dataset Track.
> [5] Seo Y, et al. State entropy maximization with random encoders for efficient exploration. ICML2021.
> [6] Mazzaglia P, et al. Curiosity-Driven Exploration via Latent Bayesian Surprise. AAAI2022.
> [7] Hansen N, et al. Temporal difference learning for model predictive control. ICML2022.
> [8] Seo Y, et al. Trajectory-wise multiple choice learning for dynamics generalization in reinforcement learning. NeurIPS2020.
> [9] Hafner D, et al. Dream to control: Learning behaviors by latent imagination. ICLR2020.

---

> ### Author Response · Authors · 2022-11-26
> **Resolving any pending concerns**
>
> We would really appreciate it if the reviewers can let us know whether they have any further concerns and whether our response addressed some/all of their concerns. We will try to address them before the discussion period ends. Thanks!

---

### Author Response · Authors · 2022-11-18
**Common Response**

We appreciate all the reviewers' valuable and inspiring comments. Individual responses and our revision has been uploaded.

For our **FIRST** revision, major updates are summarized below:

- Additional experimental results are provided, including:

    - We ablate and evaluate the role of the MCL mechanism in the complex humanoid domain in **Appendix J.1 and Figure 11 (page 20)**. Experiments show that the MCL mechanism can achieve a stable improvement in humanoid domain, which certifies the positive effect of the MCL mechanism in complex reward functions and complex environments.

    - To further verify the effectiveness of pre-training and MCL mechanism for dynamics model, we collected expert data for 30000 steps of quadruped-run tasks. Then, we train a dynamics model on the pre-collected quadruped dataset and show the prediction error curves in **Appendix J.2 and Figure 12 (page 20)** including randomly initialized, pre-trained with and without multi-choice learning using EUCLID. The results show that MCL can additionally reduce the prediction error, indirectly leading to the fast adaptation of the model.


- Add further discussion of our work from the perspective of model-based RL in **Appendix.I (red text, page 19)** (as suggested by Reviewer yoch).

- Some minor issues and unclear statements are fixed:
  - Figure legend in **Figure.1 (page 2)**
  - Only pre-training policy via diverse exploration... **(red text, page 1)**
  - Add results of TDMPC@100k baseline in **Table.1 (red text, page 8)**
  - Add clearer descriptions about how to choose the correct expert prediction head in **Appendix.G (red text, page 19)**

----------------------

For our **SECOND** revision, major updates are summarized below:

- Additional pixel-based URLB experimental results  are provided in **Appendix J.3 (page 21, revised version paper)** and Table.6, including four tasks in the pixel-based URLB (as suggested by Reviewer wBqh). The results show that EUCLID can also lead significant performance improvement in visual control tasks.  Also, we are running more experiments with different pre-training steps and trying to provide more complete results.

---------------------------------

We hope our replies have addressed all the questions and concerns the reviewers posed and shown the improved quality of the paper. **We are always willing to answer any of the reviewers' concerns about our work** and we sincerely wish the reviewers to value the technical innovation and overall contributions of the paper. We are looking forward to following inspiring discussions.

---

### Author Response · Authors · 2022-12-07
**We appreciate all reviewers' and AC's time and efforts**

We sincerely appreciate all reviewers' and AC's time and efforts in reviewing our paper. We truly thank you all for the insightful and constructive suggestions, which helped to improve the quality of our paper.

Best wishes,

Authors

---

### Decision · Program_Chairs · 2023-01-20

**Decision:**

Accept: poster

**Justification For Why Not Higher Score:**

Some issues still remain after the discussion phase:
* effective of MCL
* limited novelty
* limited contextualisation with respect to the MBRL literature

**Justification For Why Not Lower Score:**

* Combining MBRL with unsupervised RL is a promising and under-explored direction.
* Impressive results on URLB will be of interest to the community
* Acceptance will shift the discussion away from skill selection (via grid-search, or optimization)
* Highlights that the fine-tuning process could be to blame for poor performance of certain URL algorithms (i.e. DIAYN)

**Metareview: Summary, Strengths And Weaknesses:**

Reviewers agreed that the proposed method is effective, with impressive results on the unsupervised RL benchmark. Reviewers in particular appreciate the extensive experiments, inclusion of three different unsupervised RL pre-training algorithms, inclusion of the more complex humanoid domain and the headline result of monotonic improvement with pre-training (which will go a long way to making these pretraining algorithms useful in practice).

Despite this, several concerns remained after the discussion phase and virtual reviewer discussion.

(i) limited benefits of multi-choice learning. While the new results of Appendix J (combined with Table 1) give weight to the hypothesis that MCL is more effective on more complex domains, reviewers believe these fail to account for the extra parameters. Future revisions should address this point head on. Are the benefits of MCL coming from the specialized heads (is each dynamics model not a universal approximator?), from the extra diversity inducing term of Eq (5) (i.e. maintaining a diverse set of exploration policies) or simply using more parameters to model the dynamics? There is room for an extra ablation here.

(ii)  limited novelty: integrating the model-based approach of TDMPC into the unsupervised RL pre-training, with an adaptation of the multi-choice dynamics model of Seo et al., 2020 to URL;

(iii) poor contextualization with respect to the model-based RL literature.


Reviewers agree that (iii) can be addressed easily, and suggest integrating Appendix I into the main paper. As for (i) and (ii), reviewers and AC believe the benefits outweigh these particular drawbacks. At a high-level, the paper clearly shows that integrating model-based RL into the unsupervised RL pre-training phase is effective, and clearly outperforms other “skill selection” methods which have (to my knowledge) been the method of choice for fine-tuning. This will clearly be of interest to the growing unsupervised RL community, and even points to the need for more challenging benchmarks going forward. For these reasons, I believe this paper should be accepted for publication.

I nevertheless strongly urge the authors to take the feedback to heart, especially regarding points (i) and (iii). (i) should be straightforward to address, whereas (iii) will be important for this paper to be well received both by the URL and MBRL communities.

**Note From Pc:**

if the above contains the word "oral" or "spotlight" please see: "oral" presentation means -> notable-top-5% and "spotlight" means -> notable-top-25%. As stated in our emails, we are disassociating presentation type from AC recommendations

**Summary Of Ac-Reviewer Meeting:**

3/4 reviewers were present for the AC-reviewer meeting, with the exception of [E7J9] due to scheduling conflicts.

The main points of discussion and outcomes were as follows:
* We discussed the issue of novelty, vs the overall contribution of the paper. Overall most reviewers tended to agree that the paper's contribution (in terms of offering a simple solution which proves effective on the benchmark) outweighs concerns about novelty.
* We discussed whether there was sufficient empirical evidence of MCL being effective. As discussed in the meta-review, this remained a point of contention with 2/3 reviewers highlighting a potential failure mode in the ablation (lack of control for model capacity). Overall, most reviewers found that the paper could benefit from downplaying the role of MCL in favor of the "model-fused URL paradigm".
* Overall, the additional results (Appendix J), clarifications and pixel-based results were enough to convince [yoch, wBqh] to increase their score from 5 to 6.

With 4x 6's, a positive AC, and the potential for impact and interest in the community (i.e. mostly solving the URLB), all were happy to recommend acceptance.